# Learning Expensive Coordination: An Event-Based Deep RL Approach

**Runsheng Yu,**[*] **Xinrun Wang,**[*] **Rundong Wang, Youzhi Zhang, & Bo An**[†]
School of Computer Science and Engineering,
Nanyang Technological University,
Singapore
runshengyu@gmail.com,
{xwang033,rundong001,yzhang137}@e.ntu.edu.sg,
boan@ntu.edu.sg

**ZhenYu Shi**[*]**& Hanjiang Lai**[†]
School of Data and Computer Science,
Sun Yat-sen University
Guangzhou, China
shizhy6@mail2.sysu.edu.cn, laihanj3@mail.sysu.edu.cn

## Abstract

Existing works in deep Multi-Agent Reinforcement Learning (MARL) mainly focus on coordinating cooperative agents to complete certain tasks jointly. However, in many cases of the real world, agents are self-interested such as employees in a company and clubs in a league. Therefore, the leader, i.e., the manager of the company or the league, needs to provide bonuses to followers for efficient coordination, which we call *expensive coordination*. The main difficulties of expensive coordination are that i) the leader has to consider the long-term effect and predict the followers' behaviors when assigning bonuses, and ii) the complex interactions between followers make the training process hard to converge, especially when the leader's policy changes with time. In this work, we address this problem through an event-based deep RL approach. Our main contributions are threefold. (1) We model the leader's decision-making process as a semi-Markov Decision Process and propose a novel multi-agent event-based policy gradient to learn the leader's long-term policy. (2) We exploit the leader-follower consistency scheme to design a follower-aware module and a follower-specific attention module to predict the followers' behaviors and make accurate response to their behaviors. (3) We propose an action abstraction-based policy gradient algorithm to reduce the followers' decision space and thus accelerate the training process of followers. Experiments in resource collections, navigation, and the predator-prey game reveal that our approach outperforms the state-of-the-art methods dramatically.

## 1 Introduction

Deep Multi-Agent Reinforcement Learning (MARL) has been widely used in coordinating cooperative agents to jointly complete certain tasks where the agent is assumed to be selfless (fully cooperative), i.e., the agent is willing to sacrifice itself to maximize the team reward. However, in many cases of the real world, the agents are self-interested, such as taxi drivers in a taxi company (fleets) and clubs in a league. For instance, in the example of taxi fleets (Miao et al., 2016), drivers may prefer to stay in the area with high customer demand to gain more reward. It is unfair and not efficient to compel the taxi driver to selflessly contribute to the company, e.g., to stay in the low customer demand area. Forcing the drivers to selflessly contribute may increase the income for the company in a short-term but it will finally causes the low efficient and unsustainable of that company in the

---

[*]Indicates equal contribution.
[†]Co-corresponding authors.

long run because the unsatisfied drivers may be demotivated and even leave the company. Another important example is that the government wants some companies to invest on the poverty area to achieve the fairness of the society, which may inevitably reduce the profits of companies. Similar to previous example, the companies may leave when the government forces them to invest. A better way to achieve coordination among followers and achieve the leader's goals is that the manager of the company or the government needs to provide bonuses to followers, like the taxi company pays extra bonuses for serving the customers in rural areas and the government provides subsidies for investing in the poverty areas, which we term as *expensive coordination*. In this paper, we solve the large-scale sequential expensive coordination problem with a novel RL training scheme.

There are several lines of works related to the expensive coordination problem, including mechanism design (Nisan & Ronen, 2001) and the principal-agent model (Laffont & Martimort, 2009). However, these works focus more on static decisions (each agent only makes a single decision). To consider sequential decisions, the leader-follower MDP game (Sabbadin & Viet, 2013; 2016) and the RL-based mechanism design (Tang, 2017; Shen et al., 2017) are introduced but most of their works only focus on matrix games or small-scale Markov games, which *cannot* be applied to the case with the large-scale action or state space. The most related work is M³RL (Shu & Tian, 2019) where the leader assigns goals and bonuses by using a simple attention mechanism (summing/averaging the features together) and mind (behaviors) tracking to predict the followers' behaviors and makes response to the followers' behaviors. But they only consider the rule-based followers, i.e., followers with fixed preference, and ignore the followers' behaviors responding to the leader's policy, which significantly simplifies the problem and leads the unreasonability of the model.

In the expensive coordination problem, there are two critical issues which should be considered: 1) *the leader's long-term decision process* where the leader has to consider both the long-term effect of itself and long-term behaviors of the followers when determining his action to incentivise the coordination among followers, which is not considered in (Sabbadin & Viet, 2013; Mguni et al., 2019); and 2) *the complex interactions between the leader and followers* where the followers will adapt their policies to maximize their own utility given the leader's policy, which makes the training process unstable and hard, if not unable, to converge in large-scale environment, especially when the leader changes his actions frequently, which is ignored by (Tharakunnel & Bhattacharyya, 2007; Shu & Tian, 2019). In this work, we address these two issues in the expensive coordination problem through an abstraction-based deep RL approach.

Our main contributions are threefold. (1) We model the leader's decision-making process as a semi-Markov Decision Process (semi-MDP) and propose a novel event-based policy gradient to learn the leader's policy considering the long-term effect (leader takes actions at important points rather than at each step to avoid myopic decisions.) (Section 4.1). (2) A well-performing leader's policy is also highly dependent on how well the leader knows the followers. To predict the followers' behaviors precisely, we show the leader-follower consistency scheme. Based on the scheme, the follower-aware module, the follower-specific attention module, and the sequential decision module are proposed to capture these followers' behaviors and make accurate response to their behaviors (Section 4.2). (3) To accelerate the training process, we propose an action abstraction-based policy gradient algorithm for the followers. This approach is able to reduce followers' decision space and thus simplifies the interaction between the leader and followers as well as accelerates the training process of followers (Section 4.3). Experiments in resource collections, navigation and predator-prey show that our method outperforms the state-of-the-art methods dramatically.

## 2 RELATED WORKS

Our works are closely related to leader-follower RL, temporal abstraction RL, and event-based RL.

*Leader-follower RL.* The leader-follower RL targets at addressing the issue of expensive coordination where the leader wants to maximize the social benefit (or the leader's self-benefit) by coordinating non-cooperative followers through providing them bonuses. Previous works have investigated different approaches to solve the expensive coordination, including the vanilla leader-follower MARL (Sabbadin & Viet, 2013; Laumônier & Chaib-draa, 2005), leader semi-MDP (Tharakunnel & Bhattacharyya, 2007), multiple followers and sub-followers MARL (Cheng et al., 2017), followers abstraction (Sabbadin & Viet, 2016), and Bayesian optimization (Mguni et al., 2019). But most of them focus on simple tabular games or small-scale Markov games. The most related work (Shu

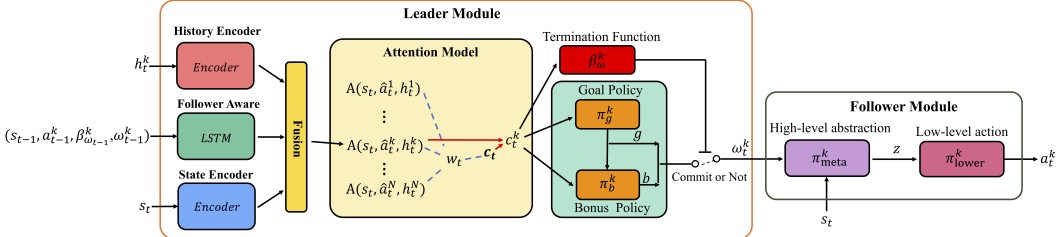

Figure 1: Overview of our framework. The details of the leader's module and the follower's module can be found in Section 4.2 and Section 4.3, respectively. The implement details of each module can be found in Appendix D.2.1.

& Tian, 2019) leverages the deep RL approach to compute the leader's policy of assigning goals and bonuses to rule-based followers. But their method performs poorly when the followers are RL-based. In this work, we aim to compute the leader's policy against the RL-based followers in the complex and sequential scenarios.

*Temporal abstraction RL.* Our methods are also related to temporal abstraction method (Sutton et al., 1998; Daniel et al., 2016; Bacon et al., 2017; Smith et al., 2018; Zhang & Whiteson, 2019; Vezhnevets et al., 2016). The basic idea of temporal abstraction is to divide the original one-level decision process into a two-level decision process where the high-level part is to decide the meta goal while the low-level policy is to select the primitive actions. Our leader's decision process is different from those methods mentioned above because the leader's policy can naturally form as an intermittent (temporal abstraction) decision process (semi-MDP) (Tharakunnel & Bhattacharyya, 2007) and it is unnecessary to design the two-level decision process for the leader (since the low-level decision process is the follower). Based on the nature of the leader, a novel training method is introduced.

*Event-based RL & Planning.* Previous studies also focus on using events to capture important elements (e.g., whether agent reaches a goal) during the whole episode. Upadhyay et al. (2018) regard the leader's action and the environment feedback as events in the continuous time environment. Becker et al. (2004); Gupta et al. (2018) leverage events to capture the fact that an agent has accomplished some goals. We adopt this idea by depicting the event as the actions taken by the leader at some time steps and design a novel event-based policy gradient to learn the long-term leader's policy.

## 3 STACKELBERG MARKOV GAMES

Our research focuses on single-leader multi-follower Stackelberg Markov Games (SMG) (Mguni et al., 2019; Sabbadin & Viet, 2013), which can be formulated as a tuple $G = \langle \mathcal{N}, \mathcal{S}, \mathcal{A}, \Omega, P, \mathcal{R}, \gamma \rangle$. $\mathcal{N}$ is the set of $N$ followers, i.e., $|\mathcal{N}| = N$. $\mathcal{S}$ is the set of states. $s_0 \in \mathcal{S}_0 \subset \mathcal{S}$ is an initial state and $S_0$ is the set of initial states. $\mathcal{A} = \times_{k \in \mathcal{N}} \mathcal{A}^k$ is the set of joint actions for followers where $a^k \in \mathcal{A}^k$ is an action for the $k$-th follower. $\boldsymbol{\omega} \in \Omega = \times_{k \in \mathcal{N}} \Omega^k$ is an action for the leader and $\omega^k = \{g^k, b^k\} \in \Omega^k$ is a goal and a bonus that the leader assigns to the $k$-th follower. $P : \mathcal{S} \times \mathcal{A} \to \Delta(\mathcal{S})$ is the transition function[1] and $\mathcal{R} = \times_{k \in \mathcal{N}} r^k \times r^l$ is the reward function set where $r^k : \mathcal{S} \times \mathcal{A} \times \Omega \to \mathbb{R}$ is the reward function for the $k$-th follower and $r^l : \mathcal{S} \times \mathcal{A} \times \Omega \to \mathbb{R}$ is the reward function for the leader. $\gamma$ is the discount factor and $\boldsymbol{a}$ is a joint action of followers.

The leader's policy is defined as $\boldsymbol{\mu} = \langle \mu^k \rangle_{k \in \mathcal{N}}$ where $\mu^k : \Omega \times \mathcal{S} \to \Delta(\Omega^k)$ is the leader's action to the $k$-th follower given the leader's action in the previous timestep $\boldsymbol{\omega}_{t-1}$ and the current state $s_t$. $\Delta(\cdot)$ is a probability distribution. The followers' joint policy is defined as $\boldsymbol{\pi} = \langle \pi^k \rangle$ where $\pi^k : \Omega^k \times \mathcal{S} \to \Delta(\mathcal{A}^k)$ is the $k$-th follower policy given the leader's action $\omega_t^k$ and the current state $s_t$. Given the policy profile of the leader and followers $\langle \boldsymbol{\mu}, \boldsymbol{\pi} \rangle$, the follower's utility is defined as $J^k(\boldsymbol{\mu}, \boldsymbol{\pi}) = \mathbb{E}\left[\sum_{t=0}^T \gamma^t r_t^k (s_t, \boldsymbol{a}_t, \boldsymbol{\omega}_t)\right]$ and the leader's utility is $J(\boldsymbol{\mu}, \boldsymbol{\pi}) = \mathbb{E}\left[\sum_{t=0}^T \gamma^t r_t^l (s_t, \boldsymbol{a}_t, \boldsymbol{\omega}_t)\right]$. We assume that the leader and followers aim to maximize their own utilities. We define the trajectory $\tau$ as a sequence of state, leader's action, and followers' actions $\langle \boldsymbol{\omega}_{-1}, (s_t, \boldsymbol{a}_t, \boldsymbol{\omega}_t)_{t=0}^T \rangle$ where $\boldsymbol{\omega}_{-1}$ is the first step leader's action and is set to zero.

---

[1]Notice that the transition function does not depend on the leader's action.

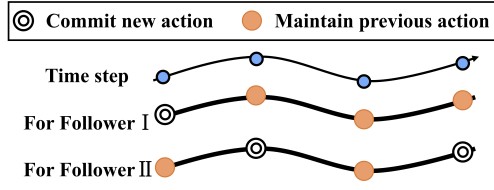
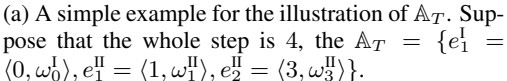

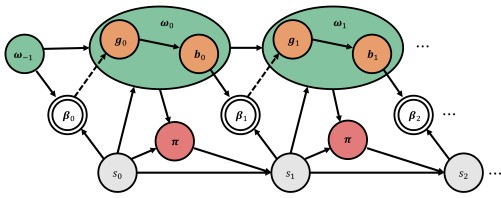

(a) A simple example for the illustration of $\mathbb{A}_T$. Suppose that the whole step is 4, the $\mathbb{A}_T = \{e_1^{\mathrm{I}} = \langle 0, \omega_0^{\mathrm{I}} \rangle, e_1^{\mathrm{II}} = \langle 1, \omega_1^{\mathrm{II}} \rangle, e_2^{\mathrm{II}} = \langle 3, \omega_3^{\mathrm{II}} \rangle \}$.

(b) The probabilistic graphical model of the proposed framework. Dotted line means that $\beta$ affects the final result of $\omega$ indirectly. $\omega_{-1}$ is set to be zero.

Figure 2: An example and a probabilistic graphical model to illustrate our method.

# 4 METHODOLOGY

In this section, we propose a novel training scheme to train a well-performing leader policy against both rule-based and RL-based followers in the expensive coordination problem. We address the two issues, the leader's long-term decision process and the complex interactions between the leader and followers, with three key steps: (a) we model the leader's decision-making process as a semi-Markov Decision Process (semi-MDP) and propose a novel event-based policy gradient to take actions only at important time steps to avoid myopic policy; (b) to accurately predict followers' behaviors, we construct a follower-aware module based on the leader-follower consistency, including a novel *follower-specific attention mechanism*, and a *sequential decision module* to predict followers' behaviors precisely and make accurate response to these behaviors; and (c) an action abstraction-based policy gradient method for followers is proposed to simplify the decision process for the followers and thus simplify the interaction between leader and followers, and accelerate the convergence of the training process.

## 4.1 EVENT-BASED TRAJECTORY OPTIMIZATION FOR LEADER

We first describe the event-based trajectory optimization for the leader. As we mentioned above, the leader's decision process can be naturally formulated as a semi-MDP (Tharakunnel & Bhattacharyya, 2007). Therefore, we firstly describe the basic ideas of semi-MDP using the modified option structure. We define the modified option as a tuple: $\langle \boldsymbol{\mu}, (\beta^k)_{k \in \mathcal{N}} \rangle$ where $\boldsymbol{\mu}$ is the leader's policy as we defined above and $\beta^k(s_t, \boldsymbol{\omega}_{t-1})$ : $\mathcal{S} \times \Omega \rightarrow [0, 1]$ is the termination function for the $k$-th follower, to indicate the probability whether the leader's action to the $k$-th follower changes ($\omega_{t-1}^k \neq \omega_t^k$). Based on these definitions, we formulate the one-step option-state transition function with decay as: $\mathrm{P}_\gamma(s_{t+1}, \boldsymbol{\omega}_t, \boldsymbol{a}_t | s_t, \boldsymbol{\omega}_{t-1}) = \gamma \mathrm{P}(s_{t+1} | s_t, \boldsymbol{a}_t) \pi(\boldsymbol{a}_t | s_t, \boldsymbol{\omega}_t) \prod_k \{(1 - \beta^k(s_t, \boldsymbol{\omega}_{t-1})) \mathbb{1}_{\omega_{t-1}^k = \omega_t^k} + \beta^k(s_t, \boldsymbol{\omega}_{t-1}) \mu^k(\omega_t^k | s_t, \boldsymbol{\omega}_{t-1})\}$, where $\mathbb{1}$ is the indicator function and $\pi(\boldsymbol{a}_t | s_t, \boldsymbol{\omega}_t) = \prod_{k \in \mathcal{N}} \pi^k(a_t^k | s_t, \omega_t^k)$ is the joint policy for followers. Notice that this is an extension of the augmented process mentioned in (Bacon et al., 2017). Differently, we do not have the low-level policy here (the low-level policy is the follower) and since we only focus on the finite time horizon, $\gamma$ is set to be 1. Our modified option is used to depict the long-term decision process for the leader as shown in Fig. 2.

Now we start to discuss our leader's policy gradient. In fact, it is not easy to directly optimize the leader's utility based on this multi-agent option-state transition function since this form includes leader's different action stages to different followers. Notice that for a sampled trajectory, *the occurrence of the leader actions* is deterministic. Therefore, we can regard the *time step* and *the action the leader takes at that step* as an event and define the (universal) event set $\mathbb{U}_T = \{\langle t_i, \omega_{t_i}^k \rangle | t_i \leq T, k \in \mathcal{N}\}$. We use the notation $e_i^k = \langle t_i, \omega_{t_i}^k \rangle$ to represent the leader's action to the $k$-th follower at step $t_i$, $i$ is the index of the event. Since we focus on the change of the actions from the leader, we further define a set that represents a collection of new actions ($\omega_t^k \neq \omega_{t-1}^k$) taken by the leader within that trajectory: $\mathbb{A}_T = \{e_i^k | \omega_{t_i}^k \neq \omega_{t_i-1}^k, t_i \leq T, k \in \mathcal{N}\} \subseteq \mathbb{U}_T$, where $t_i - 1$ is the previous time step. $\mathbb{A}_T$ represents when and how the leader commits to a new action (an example can be found in Fig. 2a). For brevity, $e_j^k \notin \mathbb{A}_T$ means $e_j^k \in \mathbb{U}_T \backslash \mathbb{A}_T$. The probability of

$\mathbb{A}_T$ can be represented as:

$$P(\mathbb{A}_T) = \prod_{k \in \mathcal{N}} \prod_{e_i^k \in \mathbb{A}_T} \beta^k(s_{t_i}, \boldsymbol{\omega}_{t_i-1}) \mu^k\left(\omega_{t_i}^k | s_{t_i}, \boldsymbol{\omega}_{t_i-1}\right) \prod_{e_j^k \notin \mathbb{A}_T} \left(1 - \beta^k(s_{t_j}, \boldsymbol{\omega}_{t_j-1})\right),$$

where $t_j - 1$ is the previous time step for $t_j$. This equation illustrates that the probability of the occurrence of a certain leader's event set within a trajectory. Concretely, the leader changes action to the $k$-th follower at $t_i \in e_i^k$ while maintaining the same action within the interval from $t_i - 1 \in e_{i-1}^k$ to $t_i$ (s.t., $t_i \in e_i^k$). Similarly, we can further define the probability of the whole trajectory $\tau$ as:

$$P(\tau) = P(s_0) \prod_{k \in \mathcal{N}} \left\{ \left[ \prod_{e_i^k \in \mathbb{A}_T} \beta^k(s_{t_i}, \boldsymbol{\omega}_{t_i-1}) \mu^k\left(\omega_{t_i}^k | s_{t_i}, \boldsymbol{\omega}_{t_i-1}\right) \pi^k\left(a_{t_i}^k | s_{t_i}, \omega_{t_i}^k\right) \right] \times \right.$$
$$\left. \left[ \prod_{e_j^k \notin \mathbb{A}_T} \left(1 - \beta^k(s_{t_j}, \boldsymbol{\omega}_{t_j-1})\right) \pi^k\left(a_{t_j}^k | s_{t_j}, \omega_{t_j}^k\right) \right] \right\} \prod_{t=0}^{T} P(s_{t+1} | s_t, \boldsymbol{a}_t).$$

Comparing with $P(\mathbb{A}_T)$, $P(\tau)$ includes the probability of the followers as well as the state transition. Do note that our goal is to maximize $\max_{\mathbb{A}_T} \mathbb{E}_{P(\tau)}\left[R^\tau(T)\right]$, indicating that the leader is required to select an action that can maximize the accumulated reward, where $R^\tau(T) = \sum_{t=0}^{T} \gamma^t r_t^l$ is the accumulated reward and $\tau$ is to stress that its accumulated reward is from the trajectory $\tau$. Following the REINFORCE trick (Sutton & Barto, 1998), the policy gradient for the termination function and the leader's policy function can be formulated under the following proposition:

**Proposition 1.** *The policy gradients for the termination function $\beta^k(s_{t_i}, \boldsymbol{\omega}_{t_i})$ and leader's policy function $\mu^k\left(\omega_{t_i}^k | s_{t_i}, \boldsymbol{\omega}_{t_i-1}\right)$ can be written as:*

$$\nabla_\theta J(\theta) \approx \mathbb{E}_{\tau \sim p_\tau(\cdot)} \left\{ \left[ \sum_{k \in \mathcal{N}} \sum_{i=0}^{T} I(e_i^k) \right] R^\tau(T) \right\}; \quad \nabla_\vartheta J(\vartheta) \approx \mathbb{E}_{\tau \sim p_\tau(\cdot)} \left\{ \left[ \sum_{k \in \mathcal{N}} \sum_{i=0}^{T} I'(e_i^k) \right] R^\tau(T) \right\};$$

*where $\theta$ and $\vartheta$ are the parameters for the termination function $\beta_\theta^k$ and leader's policy $\mu_\vartheta^k$. $I(\cdot)$ and $I'(\cdot)$ are the piece-wise functions:*

$$I(e_i^k) = \begin{cases} \dfrac{-\nabla_\theta \beta^k(s_{t_i}, \boldsymbol{\omega}_{t_i-1})}{1 - \beta^k(s_{t_i}, \boldsymbol{\omega}_{t_i-1})} & e_i^k \in \mathbb{A}_T, \\ \dfrac{\nabla_\theta \beta^k(s_{t_i}, \boldsymbol{\omega}_{t_i-1})}{\beta^k(s_{t_i}, \boldsymbol{\omega}_{t_i-1})} & e_i^k \notin \mathbb{A}_T. \end{cases} \quad I'(e_i^k) = \begin{cases} \nabla_\vartheta \log \mu^k\left(\omega_{t_i}^k | s_{t_i}, \boldsymbol{\omega}_{t_i-1}\right) & e_i^k \in \mathbb{A}_T, \\ 0 & e_i^k \notin \mathbb{A}_T. \end{cases}$$

All the proofs can be found in Appendix A. Proposition 1 implies that under the event-based method, whether the leader's commitment to a new action will induce different policy gradients for both termination function and the policy function.

However, from the empirical results, we find that the leader's policy function updates rarely during the whole episode because the policy only updates when the leader commits to a new action, which causes the sample inefficiency. Notice that in fact the leader commits to the same action when $e_i^k \notin \mathbb{A}_\mathbb{T}$. Therefore, the policy indication function $I'(\cdot)$ can be formulated in an alternative way: $I'(e_i^k) = \nabla_\vartheta \log \mu^k\left(\omega_{t_i}^k | s_{t_i}, \boldsymbol{\omega}_{t_i-1}\right), e_i^k \in \mathbb{A}_T; \nabla_\vartheta \mu^k\left(\omega_{t_i}^k = \omega_{t_i-1}^k | s_t, \boldsymbol{\omega}_{t_i-1}\right), e_i^k \notin \mathbb{A}_T$. This form considers both committing to a new action and maintaining the same actions (Details can be found in Remark 2), which we call the Event-Based Policy Gradient (EBPG) and the previous one as the sparse EBPG respectively.

Intuitively, the dense EBPG is better than the sparse EBPG because it updates the leader's policy function more frequently than the sparse one. For example, in time step $t$, supposing that the leader chooses a wrong action for follower $k$ and receives a negative reward. Then, the leader should learn to diminish the action chosen that state by EBPG. The sparse EBPG only do one PG during before terminating the action (at the committing action step) while the dense one does PG in each step before terminating the action. The latter can provide more signal to correct the wrong action. Experiments also reveal that the dense one is better (Sec. D.3.3).

### 4.2 NEURAL NETWORK BASED LEADER

The EBPG approach is able to improve leader's performance. However, it is still very hard for the leader to choose actions considering long-term effect only based on the current state information. This is because the followers change their behaviors over time according to the leader's policy.

Therefore, we introduce new modules and training schemes so as to capture the change of the followers' behaviors as well as the global state. To abstract the complicated state information, we use neural networks to learn the state representation. To capture the followers' behaviors and make accurate response to their behaviors, we design three modules: (1) we exploit the leader-follower consistency under game regularization and policy bound conditions, (2) based on the consistency, a follower-aware module is introduced and (3) based on the follower-aware module, a novel attention mechanism, and sequential decision making module is designed to make accurate response to these followers' behaviors as shown in Fig. 1.

**Leader-Follower Consistency.** In previous works, a surge of researches focus on predicting other agents' behaviors through historical information, where the other agents are assumed to be opponents of that agent, which is only suitable for zero-sum games (Zheng et al., 2018; Foerster et al., 2018; He et al., 2016). However, these methods cannot be directly applied to our case because SMG is not zero-sum. We note that Shu & Tian (2019) attempt to directly use the followers' behavior prediction module (use the history of the followers to predict their future actions) but do not analyze when and how it works. To ensure that the leader can predict the followers' behaviors, we introduce the following assumptions.

**Assumption 1.** *(Game regularization) The leader-follower state-action space ($\mathcal{A} \times \Omega \times \mathcal{S}$) is compact and $r^k$ is a continuous function w.r.t. $\boldsymbol{\mu}$ bounded by $R_{\max}$.*

This assumption is inspired by (Antos et al., 2008). We only extend it into the multi-agent forms. This assumption indicates that the action and states space should be limited and the reward function for the leader action should be smooth.

**Assumption 2.** *(Policy Bound) For any agent $k$, reward function $r^k$ and policy is consistency, i.e.,*

$$|J^k(\left[\pi^k, \pi^{-k}\right], \boldsymbol{\mu}) - J^k(\left[\pi^k, \pi^{-k}\right], \boldsymbol{\mu}')| \geq C^2 |\pi^k - \pi'^k|$$

*Where $C^2$ is a constant that satisfies $C^2 > 0$. $\pi'^k$ is the $k$-th follower's new policy. $\boldsymbol{\mu}'$ is the leader's new policy.*

This assumption is inspired by (Mguni et al., 2019). $\pi^{-k}$ indicates the joint policy without the $k$-th agent's. This assumption indicates that the change of the leader causes only slightly changes on each followers policy.

Based on these two assumptions, we propose a proposition here:

**Proposition 2.** *(Leader-Follower Consistency.) If both the assumptions of game regularization and policy bound are satisfied, for $\forall \epsilon > 0, k \in \mathcal{N}$, there exists $\delta > 0$, such that $|\boldsymbol{\mu} - \boldsymbol{\mu}'| \leq \epsilon$ implies $\left|\pi^k - \pi'^k\right| \leq \delta$, where $\boldsymbol{\mu}'$ and $\pi'^k$ are the new policies for the leader and the $k$-th follower respectively.*

This proposition reveals that the change of the leader causes only slightly changes on each follower's policy under the *game regularization* assumption and the *policy bound* assumption, which is fundamental to follower-aware learning. Roughly speaking, the *game regularization* requires that the states, actions, and rewards are bounded while the *policy bound* states that a little change of a follower's policy does not change its utility so much. The former is from the game itself and we only focus on the latter. To satisfy the latter, one possible method is to make the $\boldsymbol{\mu}$ and $\boldsymbol{\mu}'$ close because the followers always find the best response to the leader's policy and if the leader changes a little, the followers do not change too much since the new best response to $\boldsymbol{\mu}'$ is not far away from best response to $\boldsymbol{\mu}$. One direct method is to slow down the learning rate of the leader to make $\boldsymbol{\mu}'$ and $\boldsymbol{\mu}$ close. Moreover, for the leader part, taking the right actions is also an important way to guarantee the second assumption because the taken action will more precisely decrease the probability of huge change of the whole process and stabilize the training process. There is an interesting phenomenon that on one hand, knowing more about the followers can diminish the wrong decision and thus aids the establishment of the consistency. On the other hand, the consistency will further guarantee the accuracy of the follower-aware module. Therefore, they form a positive feedback.

**Follower-Aware Module.** Based on the leader-follower consistency, we can safely implement the follower-aware module to our network. Before we discuss this module in details, we first define the history for both the leader and followers. For the $k$-th follower, its history at time step $t$ is a sequence of states, its own actions, and the leader's actions to it, i.e., $h_t^k = \langle (s_{t'}, a_{t'}^k, \omega_{t'}^k)_{t' \leq t} \rangle \in \mathcal{H}_t^k$

while the leader's history is the stack of all followers histories $\boldsymbol{h}_t = \langle h_t^k \rangle_{k \in \mathcal{N}} \in \mathcal{H}_t^l$. Then, we define the history-based leader's policy as: $\mu_{\boldsymbol{h}_t}^k(s_t) = Z^{-1} p(\omega_t^k | s_t, \hat{\boldsymbol{a}}_t, \boldsymbol{h}_t) p(\hat{\boldsymbol{a}}_t | s_t, \boldsymbol{h}_t) \propto p(\omega_t^k | s_t, \hat{\boldsymbol{a}}_t, \boldsymbol{h}_t) \prod_k p^k \left( \hat{a}_t^k | s_t, h_t^k \right)$, where $Z$ is the normalization term, $p^k$ is the predicted action probability of the $k$-th follower and $\hat{a}$ defines the predicted action (predicted by the leader). $p$ is to stress the output is a probability. Since we cannot directly obtain an accurate estimation of $a^k$, we adopt an alternative way to leverage history information and imitation learning to make a prediction of other agents' action probability function $p^k \left( \hat{a}_t^k | s_t, h_t^k \right)$ (Implementation details can be found in Appendix B) and $p(\omega_t^k | s_t, \hat{\boldsymbol{a}}_t, \boldsymbol{h}_t)$ is designed using the attention mechanism as well as the sequential decision module presented below.

**Follower-Specified Attention Mechanism.** Inspired by (Chen et al., 2018), we introduce a follower-specific attention mechanism to identify the *important* followers where the important followers are followers who has just finished a task and the leader has to commit new actions to these followers. The attention mechanism is as follows:

$$w_t^k = \frac{\exp \left( f(A \left( s_t, \hat{a}_t^k, h_t^k \right)) \right)}{\sum_{k \in \mathcal{N}} \exp \left( f(A \left( s_t, \hat{a}_t^k, h_t^k \right)) \right)}; \quad \mathbf{c}_t = \sum_{k \in \mathcal{N}} w_t^k A \left( s_t, \hat{a}_t^k, h_t^k \right); c_t^k = [\mathbf{c}_t, A \left( s_t, \hat{a}_t^k, h_t^k \right)].$$

Where $w^k$ is the weight of the $k$-th follower, $A(\cdot) : \mathbb{R}^{d_s \times d_{a_t^k} \times d_{h_t^k}} \to \mathbb{R}^{d_c}$ is a function to blend various information of an agent together ($d_c$ means the dimension of the output of the $A(\cdot)$), $f(\cdot) : \mathbb{R}^{d_c} \to \mathbb{R}^1$ is a function to map the blending information to a real number, and $c_t^k$ is the $k$-th agent attention value (the compression of history, states and actions for follower $k$ as well as other followers). $\hat{a}_t^k$ is the output of $p^k \left( \hat{a}_t^k | s_t, h_t^k \right)$, the predicted the $k$-th follower's action. This attention mechanism is better because it quantifies the importance of each follower in each state through learning while the original methods only adds/averages all the features $(s_t, \hat{a}_t^k, h_t^k)$ together (Shu & Tian, 2019). Another advantage is that its weights can be visualized to see which follower is *important* to the leader at current step. (Details can be found in Appendices D.2.1 & D.3.4). $c_t^k$ then is used by $\beta^k$ and $\mu^k$.

**Sequentially Determining Goals and Bonuses.** Also notice that the goal and the bonus are sequentially correlated. Therefore, it is better for the leader to choose the bonus and the goal sequentially rather than select them independently. Therefore, to consider the goal and bonus jointly when making a decision, we build a probabilistic graph-based model as: $p(\omega_t^k | s_t, \hat{\boldsymbol{a}}_t, \boldsymbol{h}_t) \approx p(g_t^k; b_t^k | c_t^k) \propto p(b_t^k | g_t^k, c_t^k) \times p(g_t^k | c_t^k)$, the first approximate equation is established because $c_t^k$ is the compression of $(s_t, \hat{\boldsymbol{a}}_t, \boldsymbol{h}_t)$. $p(b_t^k | g_t^k, c_t^k)$ and $p(g_t^k | c_t^k)$ means the policy for bonuses and goals (Implementation details can be found in Appendix D.2.1).

### 4.3 FOLLOWER ACTION ABSTRACTION POLICY GRADIENT

These methods mentioned above are fully implemented can enhance the performance dramatically. But when facing the RL-based followers, the SMG is still hard to converge. This is because in SMG, the policies of the leader and followers are always changing depending on other agents' performance. To guarantee convergence, the leader can only update its policy when the followers reach (or are near to) the best response policy (Fiez et al., 2019). However, when the followers are RL-based agents, there is no way to ensure the followers' policies are (near) the best response policies in large-scale SMG and the commonly-seen idea is to provide enough training time but it is unbearable in practice due to the limitation of computing power (Mguni et al., 2019).

To accelerate the training process, inspired by the action abstraction approach which is commonly-seen in Poker (Brown & Sandholm, 2019; Tuyls et al., 2018) and action abstraction RL (Chandak et al., 2019), we collect the followers' primitive actions sharing the same properties together as a meta policy. Then, the followers only need to select the meta action to make a decision. Therefore, the original game is converted into a meta game, which is easy to solve.

Specifically, we define the policy for the $k$-th follower as: $\pi_t^k(a^k | \hat{s}) = \sum_z \pi_{meta}^k(z | \hat{s}) \pi_{lower}^k(a | \hat{s}, z)$, where $\hat{s} = \langle s, \omega^k \rangle$ is the augmented state for the follower (the combination of current state and the leader's action). $\pi_{meta}^k(z | \hat{s})$ is the meta policy for the $k$-th follower and $z$ is the high-level (meta) action. We hypothesize that the lower-level policy (the policy to choose the primitive actions) is already known (rule-based) and deterministic, i.e., $\pi_{lower}^k(a^k | \hat{s}, z) = 1$. For instance, given the example of the navigation task, the $\pi_{meta}^k$ can be the selection to which landmark to explore while $\pi_{lower}^k$ is a specific route planning algorithm (such

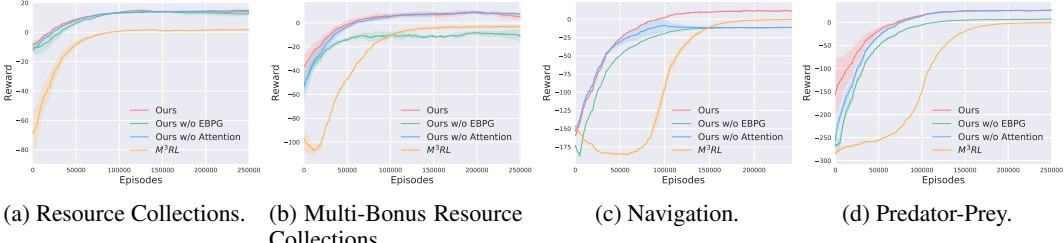

(a) Resource Collections.   (b) Multi-Bonus Resource   (c) Navigation.   (d) Predator-Prey.
                             Collections.

Figure 3: Leader's reward curves for different tasks (rule-based followers).

as Dijkstra Algorithm). Based on this assumption, we can design a novel policy gradient to train the meta policy: $\nabla_{\lambda^k} J^k \propto \mathbb{E}\left[\nabla_{\lambda_k} \log \pi^k_{meta}(z|\hat{s})R^k\right]$, where $\lambda^k$ is the parameter for meta-policy $\pi^k_{meta}$ (Details can be found in Lemma 3).

### 4.4 LOSS FUNCTIONS

In this section, we discuss how to design the leader's and followers' loss functions.

**Loss Functions for the Leaders.** The basic structure for the leader is the actor-critic structure (Sutton & Barto, 1998). We find that adding regularizers can enhance the leader's performance and we implement the maximum entropy for the leader's policy function as well as the L2 regularization for the termination function, i.e., $L_{enp} = -\sum_k \sum_{\omega^k} \mu^k(\omega^k|s, \boldsymbol{h}) \log \mu^k(\omega^k|s, \boldsymbol{h})$ and $L_{reg} = \beta^2$. We also use imitation learning to learn the predicted action function $p^k$. Following the same logic of (Shu & Tian, 2019), two baseline functions $\phi^g(c_t)$ and $\phi^b(c_t)$ are also introduced to further reduce the variance. Details can be found in Appendix B.

**Loss Functions for the RL-Based Followers.** The basic structure for each follower is also based on the actor-critic structure. We leverage the action abstraction policy gradient as we mentioned above. The learning rate between the leader and follower should satisfy the two time-scale principle (Roughly speaking, the leader learns slower than the follower(s)), similar to (Borkar, 1997). Details can be found in Appendix B and the pseudo-code can be found in Appendix C.

## 5 EXPERIMENTAL RESULTS

### 5.1 SETUP

**Tasks.** We evaluate the following tasks to testify the performance of our proposed method. All of these tasks are based on SMG mentioned above. (1) resource collections: each follower collects three types of resources including its preferred one and the leader can choose two bonuses levels (Shu & Tian, 2019); (2) multi-bonus resources collections: based on (1), the leader can choose four bonuses levels; (3) modified navigation: followers are required to navigate some landmarks and after one of the landmarks is reached, the reached landmark disappears and new landmark will appear randomly. (4) modified predator-prey: followers are required to capture some randomly moving preys, prizes will be given after touching them. Both (3) and (4) are based on (Lowe et al., 2017) and we modify them into our SMG setting. Moreover, to increase the difficulty, in each episode, the combinations of the followers will change, i.e., in each task, there are 40 different followers and at each episode, we randomly choose some followers to play the game. More details can be found in Appendix D.

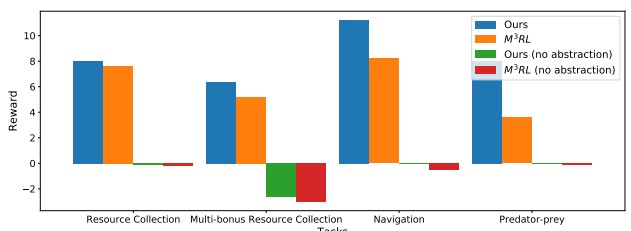

Figure 4: The final reward for RL-based followers. No abstraction means the vanilla RL-based followers.

**Baselines & Ablations.** To evaluate our method, we compare a recently proposed method as our baseline: M³RL (Shu & Tian, 2019). We do not include other baselines because other methods

cannot be used in our problems, as justified in (Shu & Tian, 2019). For the ablations of the leader part, we choose: (1) ours: the full implementation of our method. (2) ours w/o EBPG: removing the event-based policy gradient part; (3) ours w/o Attention: replacing follower-specified attention model by the original attention model mentioned in (Shu & Tian, 2019). For the follower part, we choose (a) with rule-based follower (b) with vanilla RL-based follower, and (c) with action abstraction RL-based follower to testify the ability of our methods when facing different followers.

**Hyper-Parameters.** Our code is implemented in Pytorch (Paszke et al., 2017). If no special mention, the batch size is 1 (online learning). Similar to (Shu & Tian, 2019), we set the learning rate as 0.001 for the leader's critic and followers while 0.0003 for the leader's policy. The optimization algorithm is Adam (Kingma & Ba, 2014). Our method takes less than two days to train on a NVIDIA Geforce GTX 1080Ti GPU in each experiment.

For the loss function, we set the $\lambda_1 = 0.01$ and $\lambda_2 = 0.001$. The total training episode is $250,000$ for all the tasks (including both the rule-based followers and the RL-based followers). To encourage exploration, we use the $\iota$-greedy[2]. For the leader, the exploration rate is set to $0.1$ and slightly decreases to zero (5000 episode). For the followers, the exploration rate for each agent is always $0.3$ (except for the noise experiments).

## 5.2 LEARNING EFFICIENCY

The quantitative results with different tasks are shown in Figs. 3 & 4. For the *rule-based followers*, from Fig. 3, we find that our method outperforms the state-of-the-art method in all the tasks, showing that our method

Table 1: Robustness results in multi-bound resource collections. $b\%$ is the probability that followers randomly choose actions.

| Methods | 0% Noise | 30% Noise | 50% Noise |
|---|---|---|---|
| Ours total incentive | **18.32** | **17.63** | **17.28** |
| M³RL total incentive | 4.06 | 3.85 | 4.02 |
| Ours total reward | **10.06** | **5.36** | **5.30** |
| M³RL total reward | -1.58 | -3.23 | -8.96 |

is sample efficient and fast to coverage. There is an interesting phenomenon that in the task of multi-bonus resource collections and navigation, only our method obtains a positive reward, indicating that our method can work well in complicated environments. For ablations, we can see that ours w/o attention and ours w/o EBPG are worse than ours, representing these components do enhance the performance. For the *RL-based followers*, from Fig. 4, we observe that when facing the RL-based method with action abstraction, our approach outperforms the baseline method in all the tasks (in predator-prey game, the reward for ours is twice as that of the state-of-the-art). We also find that without action abstraction, the reward is less than *zero*, revealing that the abstraction does play a crucial role in stabilizing training.

## 5.3 ROBUSTNESS

This experiment is to evaluate whether our method is robust to the noise, i.e., the follower randomly takes actions. We make this experiment by introducing noise into the follower decision. From Table 1, we can find that our method reaches a higher total reward (more than 5) among all the environment with noise than the state-of-the-art, indicating that our method is robust to the noise. We also observe that the total reward for the baseline method becomes lower with the increase of the noise while our method is more robust to the change. Moreover, for the incentive (the total gain), we find that our method gains much more incentive than the state-of-the-art method, showing that our method coordinates have a better coordination the followers than the state-of-the-art method.

## 5.4 MORE EXPERIMENTS

We also do a substantial number of experiments. However, due to the space limitation, we can only provide some results here: (1) The total incentives: incentive can reveal the performance of successful rate interacting with the followers. Our method outperforms the state-of-the-art method, indicating that our method has a better ability to interact with the followers. (2) Sparse EBPG: we compare the performance gap between sparse EBPG and (dense) EBPG. This results show that the sparse one is worse than the dense one, supporting the assumption that the dense signal can improve

---

[2]Normally it is called the decayed $\epsilon$-greedy. We use $\iota$ instead of $\epsilon$ to avoid notation abuse.

the sample efficiency. (3) Visualizing attention: We visualize the attention module to find what it actually learns and the result indicates that our attention mechanism does capture the followers whom the leader needs to assign bonuses to. (4) Two time-scale training: We testify whether our two time-scale training scheme works and the ablation shows that this scheme does play an important role in improving the performance of both the leader and the followers. (5) The committing interval: We observe that the dynamic committing interval (our method) performs better than the one with fixed committing intervals. (6) Reward for RL-based followers: we show the reward for the followers, which can provide the situation of the followers. The result represents that our method aids the followers to gain more than the state-of-the-art method. (7) Number of RL-based followers: finally, we testify our method in cases with different number of RL-based followers. The result shows that our method always performs well. The full results can be found in Appendix D.

## 6 CONCLUSION REMARKS

This paper proposes a novel RL training scheme for Stackelberg Markov Games with single leader and multiple self-interested followers, which considers the leader's long-term decision process and complicated interaction between followers with three contributions. 1) To consider the long-term effect of the leader's behavior, we develop an event-based policy gradient for the leader's policy. 2) To predict the followers' behaviors and make accurate response to their behaviors, we exploit the leader-follower consistency to design a novel follower-aware module and follower-specific attention mechanism. 3) We propose an action abstraction-based policy gradient algorithm to accelerate the training process of followers. Experiments in resource collections, navigation, and predator-prey game reveal that our method outperforms the state-of-the-art methods dramatically.

We are willing to highlight that SMGs contribute to the RL (especially MARL) community with three key aspects: 1). As we mentioned in the Introduction, most of the existing MARL methods assume that all the agents are willing to sacrifice themselves to maximize the total rewards, which is not true in many real-world non-cooperative scenarios. On the contrary, our proposed method realistically assumes that agents are self-interested. Thus, SMGs provide a new scheme focusing more on the self-interested agents. We think this aspect is the most significant contribution to the RL community. 2). The SMGs can be regarded as the multi-agent system with different roles (the leader and the followers) (Wilson et al., 2008) and our method provides a solution to that problem. 3). Our methods also contribute to the hierarchical RL, i.e., it provides a non-cooperative training scheme between the high-level policy (the leaders) and the low-level policy (the followers), which plays an important role when the followers are self-interested. Moreover, our EBPG also propose an novel policy gradient method for the temporal abstraction structure.

There are several directions we would like to investigate to further extend our SMG model: i) we will consider multiple cooperative/competitive leaders and multiple self-interested followers, which is the case in the labor market, ii) we will consider multi-level leaders, which is the case in the hierarchical organizations and companies and iii) we will consider the adversarial attacks to our SMG model, which may induce extra cost to the leader for efficient coordination. We believe that our work is a preliminary step towards a deeper understanding of the leader-follower scheme in both research and the application to society.

### ACKNOWLEDGEMENTS

This research is supported by NRF AISG-RP-2019-0013, NSOE-TSS2019-01, MOE and NTU.

Also, this work is supported by the National Natural Science Foundation of China under Grants (U1611264, U1811261,61602530, 61772567, U1811262 and U1711262). This work is also supported by the Pearl River Nova Program of Guangzhou (201906010080).

We would like to thank Tianming Shu, Darren Chua, Suming Yu, Enrique Munoz de Cote, and Xu He for their kind suggestions and helps. We also appreciate the anonymous reviewers for their useful suggestions.

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

## A  PROOFS

**Proposition 1.** *The policy gradients for termination function $\beta^k\left(s_{t_i}, \boldsymbol{\omega}_{t_i}\right)$ and leader's policy function $\mu^k\left(\omega_{t_i}^k | s_{t_i}, \boldsymbol{\omega}_{t_i-1}\right)$ can be written as:*

$$\nabla_\theta J(\theta) \approx \mathbb{E}_{\tau \sim p_\tau(\cdot)}\left\{\left[\sum_{k \in \mathcal{N}} \sum_{i=0}^{T} I(e_i^k)\right] R^\tau(T)\right\}; \quad \nabla_\vartheta J(\vartheta) \approx \mathbb{E}_{\tau \sim p_\tau(\cdot)}\left\{\left[\sum_{k \in \mathcal{N}} \sum_{i=0}^{T} I'(e_i^k)\right] R^\tau(T)\right\};$$

*where $\theta$ and $\vartheta$ are the parameters for the termination function $\beta_\theta^k$ and the leader's policy $\mu_\vartheta^k$. $I(\cdot)$ and $I'(\cdot)$ are the piece-wise functions:*

$$I(e_i^k) = \begin{cases} \frac{-\nabla_\theta \beta^k\left(s_{t_i}, \boldsymbol{\omega}_{t_i-1}\right)}{1-\beta^k\left(s_{t_i}, \boldsymbol{\omega}_{t_i-1}\right)} & e_i^k \in \mathbb{A}_T \\ \frac{\nabla_\theta \beta^k\left(s_{t_i}, \boldsymbol{\omega}_{t_i-1}\right)}{\beta^k\left(s_{t_i}, \boldsymbol{\omega}_{t_i-1}\right)} & e_i^k \notin \mathbb{A}_T. \end{cases} \quad I'(e_i^k) = \begin{cases} \nabla_\vartheta \log \mu^k\left(\omega_{t_i}^k | s_{t_i}, \boldsymbol{\omega}_{t_i-1}\right) & e_i^k \in \mathbb{A}_T \\ 0 & e_i^k \notin \mathbb{A}_T \end{cases}$$

*Proof.* First recall the utility for the leader:

$$J(\theta) = \mathbb{E}_{P(\tau)}\left[R^*(T)\right]$$

$$= \sum_\tau \sum_m \mathcal{P}(|\mathbb{A}_T| = m) P(s_0) \prod_{k \in \mathcal{N}} \left\{\left[\prod_{e_i^k \in \mathbb{A}_T} \beta^k(s_{t_i}, \boldsymbol{\omega}_{t_i-1}) \mu^k\left(\omega_{t_i}^k | s_{t_i}, \boldsymbol{\omega}_{t_i-1}\right) \pi^k\left(a_{t_i}^k | s_{t_i}, \omega_{t_i}^k\right)\right] \times \right.$$

$$\left.\left[\prod_{e_j^k \notin \mathbb{A}_T} \left(1 - \beta^k(s_{t_j}, \boldsymbol{\omega}_{t_j-1})\right) \pi^k\left(a_{t_j}^k | s_{t_j}, \omega_{t_j}^k\right)\right]\right\} \prod_{t=0}^{T} P(s_{t+1} | s_t, \boldsymbol{a}_t) R^\tau(T);$$

Where $m$ is the number of the times taking new action, $m \leq T$. If $m = T$, implying that the leader has taken new action at each time. $\mathcal{P}(|\mathbb{A}_T| = m)$ means the probability of times taking new action within an episode.

Take derivatives on both LHS and RHS, we get:

$$\nabla_\theta J(\theta) = \sum_\tau \sum_m \nabla_\theta \mathcal{P}(|\mathbb{A}_T| = m) P(s_0) \prod_{k \in \mathcal{N}} \left\{\left[\prod_{e_i^k \in \mathbb{A}_T} \beta^k(s_{t_i}, \boldsymbol{\omega}_{t_i-1}) \mu^k\left(\omega_{t_i}^k | s_{t_i}, \boldsymbol{\omega}_{t_i-1}\right) \pi^k\left(a_{t_i}^k | s_{t_i}, \omega_{t_i}^k\right)\right] \times \right.$$

$$\left.\left[\prod_{e_j^k \notin \mathbb{A}_T} \left(1 - \beta^k(s_{t_j}, \boldsymbol{\omega}_{t_j-1})\right) \pi^k\left(a_{t_j}^k | s_{t_j}, \omega_{t_j}^k\right)\right]\right\} \prod_{t=0}^{T} P(s_{t+1} | s_t, \boldsymbol{a}_t) R^\tau(T)$$

$$= \sum_\tau \sum_m \frac{\nabla_\theta \mathcal{P}(|\mathbb{A}_T| = m) \prod_{k \in \mathcal{N}} \prod_{e_i^k \in \mathbb{A}_T} \beta^k(s_{t_i}, \boldsymbol{\omega}_{t_i-1}) \prod_{e_j^k \notin \mathbb{A}_T} (1 - \beta^k(s_{t_j}, \boldsymbol{\omega}_{t_j-1}))}{\mathcal{P}(|\mathbb{A}_T| = m) \prod_{k \in \mathcal{N}} \prod_{e_i^k \in \mathbb{A}_T} \beta^k(s_{t_i}, \boldsymbol{\omega}_{t_i-1}) \prod_{e_j^k \notin \mathbb{A}_T} (1 - \beta^k(s_{t_j}, \boldsymbol{\omega}_{t_j-1}))}$$

$$\times \mathcal{P}(|\mathbb{A}_T| = m) P(s_0) \prod_{k \in \mathcal{N}} \left\{\left[\prod_{e_i^k \in \mathbb{A}_T} \beta^k(s_{t_i}, \boldsymbol{\omega}_{t_i-1}) \mu^k\left(\omega_{t_i}^k | s_{t_i}, \boldsymbol{\omega}_{t_i-1}\right) \pi^k\left(a_{t_i}^k | s_{t_i}, \omega_{t_i}^k\right)\right] \times \right.$$

$$\left.\left[\prod_{e_j^k \notin \mathbb{A}_T} \left(1 - \beta^k(s_{t_j}, \boldsymbol{\omega}_{t_j-1})\right) \pi^k\left(a_{t_j}^k | s_{t_j}, \omega_{t_j}^k\right)\right]\right\} \prod_{t=0}^{T} P(s_{t+1} | s_t, \boldsymbol{a}_t) R^\tau(T)$$

For brevity, we use $P(\tau)$ to represent $P(s_0) \prod_{k \in \mathcal{N}} \left\{\left[\prod_{e_i^k \in \mathbb{A}_T} \beta^k(s_{t_i}, \boldsymbol{\omega}_{t_i-1}) \mu^k\left(\omega_{t_i}^k | s_{t_i}, \boldsymbol{\omega}_{t_i-1}\right) \pi^k\left(a_{t_i}^k | s_{t_i}, \omega_{t_i}^k\right)\right] \times \right.$ $\left.\left[\prod_{e_j^k \notin \mathbb{A}_T} \left(1 - \beta^k(s_{t_j}, \boldsymbol{\omega}_{t_j-1})\right) \pi^k\left(a_{t_j}^k | s_{t_j}, \omega_{t_j}^k\right)\right]\right\} \prod_{t=0}^{T} P(s_{t+1} | s_t, \boldsymbol{a}_t)$, the trajectory probability. And we use $i \in e_i^k$ and $j \in e_j^k$ to represent $i \in e_i^k \in \mathbb{A}_T$ to $j \in e_j^k \notin \mathbb{A}_T$ with a slight abuse of

notation. Thus the equation mentioned above can be further simplified as:

$$= \sum_{\tau} \sum_{m} \mathcal{P}(|\mathbb{A}_T| = m) \nabla_\theta \left( \sum_{k\in\mathcal{N}} \sum_{i\in e_i^k} \log(1 - \beta^k(s_{t_i}, \boldsymbol{\omega}_{t_i-1})) + \sum_{k\in\mathcal{N}} \sum_{j\in e_j^k} \log \beta^k(s_{t_j}, \boldsymbol{\omega}_{t_j-1}) \right) P(\tau) R^\tau(T)$$

$$\approx \mathbb{E}_{\tau\sim p_\tau(\cdot)} \left\{ \nabla_\theta \left[ \sum_{k\in\mathcal{N}} \sum_{i\in e_i^k} \log(1 - \beta^k(s_{t_i}, \boldsymbol{\omega}_{t_i-1})) + \sum_{k\in\mathcal{N}} \sum_{j\in e_j^k} \log \beta^k(s_{t_j}, \boldsymbol{\omega}_{t_j-1}) \right] R^\tau(T) \right\}$$

$$= \mathbb{E}_{\tau\sim p_\tau(\cdot)} \left\{ \left[ \sum_{k\in\mathcal{N}} \sum_{i\in e_i^k} \frac{-\nabla_\theta \beta^k(s_{t_i}, \boldsymbol{\omega}_{t_i-1})}{1 - \beta^k(s_{t_i}, \boldsymbol{\omega}_{t_i-1})} + \sum_{k\in\mathcal{N}} \sum_{j\in e_j^k} \frac{\nabla_\theta \beta^k(s_{t_j}, \boldsymbol{\omega}_{t_j-1})}{\beta^k(s_{t_j}, \boldsymbol{\omega}_{t_j-1})} \right] R^\tau(T) \right\}$$

The equation above is exactly the REINFORCE trick (Sutton & Barto, 1998) and the rule of derivations. The approximation indicates that one trajectory only has one $\mathbb{A}_T$[3]. Also based on the definition of $e_i^k$ and $e_j^k$, the equation can be rewritten in a more compact form:

$$\nabla_\theta J(\theta) = \mathbb{E}_{\tau\sim p_\tau(\cdot)} \left\{ \left[ \sum_{k\in\mathcal{N}} \sum_{i\in e_i^k} \frac{-\nabla_\theta \beta^k(s_{t_i}, \boldsymbol{\omega}_{t_i-1})}{1 - \beta^k(s_{t_i}, \boldsymbol{\omega}_{t_i-1})} + \sum_{k\in\mathcal{N}} \sum_{j\in e_j^k} \frac{\nabla_\theta \beta^k(s_{t_j}, \boldsymbol{\omega}_{t_j-1})}{\beta^k(s_{t_j}, \boldsymbol{\omega}_{t_j-1})} \right] R^\tau(T) \right\}$$

$$= \mathbb{E}_{\tau\sim p_\tau(\cdot)} \left\{ \left[ \sum_{k\in\mathcal{N}} \sum_{i=0}^{T} I(e_i^k) \right] R^\tau(T) \right\},$$

Where $I(\cdot)$ is the piece-wise function:

$$I(e_i^k) = \begin{cases} \frac{-\nabla_\theta \beta^k(s_{t_i}, \boldsymbol{\omega}_{t_i-1})}{1 - \beta^k(s_{t_i}, \boldsymbol{\omega}_{t_i-1})} & e_i^k \in \mathbb{A}_T \\ \frac{\nabla_\theta \beta^k(s_{t_i}, \boldsymbol{\omega}_{t_i-1})}{\beta^k(s_{t_i}, \boldsymbol{\omega}_{t_i-1})} & e_i^k \notin \mathbb{A}_T. \end{cases}$$

This is the first part of the proof (the policy gradient for the termination function). Here, we start proving the second part (the policy gradient for the leader's action).

The proof of the second part is similar to the first part.

$$\nabla_\vartheta J(\vartheta) = \sum_{\tau} \sum_{m} \mathcal{P}(|\mathbb{A}_T| = m) P(s_0) \prod_{k\in\mathcal{N}} \left\{ \left[ \prod_{e_i^k \in \mathbb{A}_T} \beta^k(s_{t_i}, \boldsymbol{\omega}_{t_i-1}) \mu^k(\omega_{t_i}^k | s_{t_i}, \boldsymbol{\omega}_{t_i-1}) \pi^k(a_{t_i}^k | s_{t_i}, \omega_{t_i}^k) \right] \times$$

$$\left[ \prod_{e_j^k \notin \mathbb{A}_T} (1 - \beta^k(s_{t_j}, \boldsymbol{\omega}_{t_j-1})) \right] \right\} \prod_{t=0}^{T} P(s_{t+1} | s_t, \boldsymbol{a}_t) R^\tau(T);$$

$$= \sum_{\tau} \sum_{m} \frac{\nabla_\vartheta \mathcal{P}(|\mathbb{A}_T| = m) \prod_{k\in\mathcal{N}} \prod_{e_i^k \in \mathbb{A}_T} \mu^k(\omega_{t_i}^k | s_{t_i}, \boldsymbol{\omega}_{t_i-1})}{P(|\mathbb{A}_T| = m) \prod_{k\in\mathcal{N}} \prod_{e_i^k \in \mathbb{A}_T} \mu^k(\omega_{t_i}^k | s_{t_i}, \boldsymbol{\omega}_{t_i-1})}$$

$$\times \mathcal{P}(|\mathbb{A}_T| = m) \times P(s_0) \prod_{k\in\mathcal{N}} \prod_{i\in e_i^k} \beta^k(s_{t_i}, \boldsymbol{\omega}_{t_i-1}) \mu^k(\omega_{t_i}^k | s_{t_i}, \boldsymbol{\omega}_{t_i-1}) \pi^k(a_{t_i}^k | s_{t_i}, \omega_{t_i}^k)$$

$$\times \prod_{e_j^k \notin \mathbb{A}_T} (1 - \beta^k(s_{t_j}, \boldsymbol{\omega}_{t_j-1})) \pi^k(a_{t_j}^k | s_{t_j}, \omega_{t_j}^k) R^\tau(T)$$

$$= \sum_{\tau} \sum_{m} \left( \sum_{k\in\mathcal{N}} \sum_{e_i^k} \nabla_\vartheta \log \mu^k(\omega_{t_i}^k | s_{t_i}, \boldsymbol{\omega}_{t_i-1}) \right) P(\tau) R^\tau(T)$$

$$= \mathbb{E}_{\tau\sim p_\tau(\cdot)} \left\{ \left[ \sum_{k\in\mathcal{N}} \sum_{i\in e_i^k} \nabla_\vartheta \log \mu^k(\omega_{t_i}^k | s_{t_i}, \boldsymbol{\omega}_{t_i-1}) \right] R^\tau(T) \right\}$$

---

[3]We find that Upadhyay et al. (2018) also implement this approximation but use different explanations.

We rewrite it to a more compact form:

$$\nabla_\vartheta J(\vartheta) = \mathbb{E}_{\tau \sim p_\tau(\cdot)} \left\{ \left[ \sum_{k \in \mathcal{N}} \sum_{i=1}^{T} I'(e_t^k) \right] R^\tau(T) \right\}$$

$$I'(e_i^k) = \begin{cases} \nabla_\vartheta \log \mu^k \left( \omega_{t_i}^k | s_{t_i}, \boldsymbol{\omega}_{t_i-1} \right) & e_i^k \in \mathbb{A}_T \\ 0 & e_i^k \notin \mathbb{A}_T \end{cases}$$

$\square$

**Remark 1.** *Some researches also focus on event-based RL but either on single-agent continuous time (Upadhyay et al., 2018) or reward representation (Gupta et al., 2018). We are the first to develop and implement the event-based policy gradient into the multi-agent system.*

**Remark 2.** *In fact, the policy gradient for the leader actions might be somewhat sparse, i.e., we only update the policy when the leader changes its actions. Notice that the leader commits to the same action when $e_i^k \notin \mathbb{A}_\mathbb{T}$. Therefore, the probability of leader's action $P(\mathbb{A}_\mathbb{T})$ can also represented as:*

$$P(\mathbb{A}_\mathbb{T}) = \prod_{k \in \mathcal{N}} \prod_{e_i^k \in \mathbb{A}_T} \beta^k(s_{t_i}, \boldsymbol{\omega}_{t_i-1}) \mu^k \left( \omega_{t_i}^k | s_{t_i}, \boldsymbol{\omega}_{t_i-1} \right)$$

$$\times \prod_{e_j^k \notin \mathbb{A}_T} \left( 1 - \beta^k(s_{t_j}, \boldsymbol{\omega}_{t_j-1}) \right) \mu^k \left( \omega_{t_j}^k = \omega_{t_j-1}^k | s_{t_j}, \boldsymbol{\omega}_{t_j-1} \right);$$

*Then the policy gradient for leader's policy $\nabla_\vartheta J(\vartheta)$ can thus be $\nabla_\vartheta J(\vartheta) \approx \mathbb{E}_{\tau \sim p_\tau(\cdot)} \left\{ \left[ \sum_{k \in \mathcal{N}} \sum_{i=0}^{T} I'(e_i^k) \right] R^\tau(T) \right\}$, where*

$$I'(e_i^k) = \begin{cases} \nabla_\vartheta \log \mu^k \left( \omega_{t_i}^k | s_{t_i}, \boldsymbol{\omega}_{t_i-1} \right) & e_i^k \in \mathbb{A}_T \\ \nabla_\vartheta \mu^k \left( \omega_{t_i}^k = \omega_{t_i-1}^k | s_t, \boldsymbol{\omega}_{t_i-1} \right) & e_i^k \notin \mathbb{A}_T \end{cases}$$

**Lemma 1.** *(Reward Bound) For any agent $k$, the corresponding reward function $r^k$ w.r.t $\omega$ is $C$-Lipschitz continuous.*

$$|r_t^k \left( s_t, a_t^k, a_t^{-k}, \boldsymbol{\mu} \right) - r_t^k \left( s_t, a_t^k, a_t^{-k}, \boldsymbol{\mu}' \right)| \leq C|\boldsymbol{\mu} - \boldsymbol{\mu}'|, s_t, \boldsymbol{a}_t = [a_t^k, a_t^{-k}].$$

*Where $C$ is a constant that satisfies $C > 0$.[4] $\boldsymbol{\mu}'$ is leader's new policy. $a^{-k}$ is the joint action without agent $k$'s.*

*Proof.* Based on Assumption 1, we can build a compact metric space $(\mathcal{A} \times \Omega \times \mathcal{S}, R^k)$. From the Heine-Cantor theorem we know that the compact metric space induce uniformly continuous.

That is, for every $\epsilon > 0$, there exists a $\delta > 0$, such that $|\boldsymbol{\mu} - \boldsymbol{\mu}'| \leq \epsilon$ implies $|r_t^k \left( s_t, a_t^k, a_t^{-k}, \boldsymbol{\mu} \right) - r_t^k \left( s_t, a_t^k, a_t^{-k}, \boldsymbol{\mu}' \right)| \leq \delta$. There exists at least one positive Constance $C$ such that $\delta \leq C/\epsilon$. Then $|r_t^k \left( s_t, a_t^k, a_t^{-k}, \boldsymbol{\mu} \right) - r_t^k \left( s_t, a_t^k, a_t^{-k}, \boldsymbol{\mu}' \right)| \leq \delta \leq \frac{C}{\epsilon} \leq C|\boldsymbol{\mu} - \boldsymbol{\mu}'|$. $\square$

This Lemma is similar to the assumption in (Mguni et al., 2019). We prove it rather than make it an assumption.

**Lemma 2.** *If Assumption 1 is satisfied, the inequality is established:*

$$|J^k([\pi^k, \pi^{-k}], \boldsymbol{\mu}) - J^k([\pi^k, \pi^{-k}], \boldsymbol{\mu}')| \leq \mathcal{C}|\boldsymbol{\mu} - \boldsymbol{\mu}'|$$

*Proof.* Expand the utility function by the bellman equation:

---

[4] The follower's reward is originally defined based on the $r_t^k \left( s_t, a_t^k, a_t^{-k}, \boldsymbol{\omega}_t \sim \boldsymbol{\mu} \right)$. Here to emphasize the relationship between leader's policy $\boldsymbol{\mu}$ and reward, with a little abuse of the notation, we use $r_t^k \left( s_t, a_t^k, a_t^{-k}, \boldsymbol{\mu} \right)$ to represent the reward function.

$$\left| J^k \left( \left[ \pi^k, \pi^{-k} \right], \boldsymbol{\mu} \right) - J^k \left( \left[ \pi^k, \pi^{-k} \right], \boldsymbol{\mu}' \right) \right| \tag{1}$$

$$= | \max_{\boldsymbol{\pi} \in \Pi} \mathbb{E} \left[ r^k \left( s_0, a_0^k, a_0^{-k}, \boldsymbol{\mu} \right) + \gamma \sum_{s_1 \in S} P \left( s_1 | s_0, \boldsymbol{a}_0 \right) V_1^k \left( \left[ \pi_1^k, \pi_1^{-k} \right], \boldsymbol{\mu}, s_1 \right) \right]$$

$$- \max_{\boldsymbol{\pi} \in \Pi} \mathbb{E} \left[ r^k \left( s_0, a_0^k, a_0^{-k}, \boldsymbol{\mu}' \right) + \gamma \sum_{s_1 \in S} P \left( s_1 | s_0, \boldsymbol{a}_0 \right) V_1^k ( \left[ \pi_1^k, \pi_1^{-k} \right], \boldsymbol{\mu}', s_1) \right] | \tag{2}$$

$$\leq \max_{\boldsymbol{\pi} \in \Pi} | \mathbb{E} \{ r^k \left( s_0, a_0^k, a_0^{-k}, \boldsymbol{\mu} \right) - r^k \left( s_0, a_0^k, a_0^{-k}, \boldsymbol{\mu}' \right)$$

$$+ \gamma \sum_{s_1 \in S} P \left( s' | s, \boldsymbol{a} \right) \left[ V_1^k ( \left[ \pi_1^k, \pi_1^{-k} \right], \boldsymbol{\mu}, s_1) - V_1^k ( \left[ \pi_1^k, \pi_1^{-k} \right], \boldsymbol{\mu}', s_1) \right] \} |, \tag{3}$$

$$= \sum_{s \in S} \sum_{t=0}^{\infty} \gamma^t P^{\boldsymbol{\pi}} \left( s_{t+1} | s_t, \boldsymbol{\pi} \right) \max_{\boldsymbol{\pi} \in \Pi} \left| \mathbb{E} \left[ r^k \left( s_t, a_t^k, a_t^{-k}, \boldsymbol{\mu} \right) - r^k \left( s_t, a_t^k, a_t^{-k}, \boldsymbol{\mu}' \right) \right] \right| \tag{4}$$

The first relaxation is due to the property of max: $\max | A(x) - B(x) | \geq | \max A(x) - \max B(x) |$, where $A(x)$ and $B(x)$ are both the real functions. The Eq. (4) is the result of recursive iteration.

$$\leq (1 - \gamma)^{-1} \max_{\boldsymbol{\pi} \in \Pi} | \mathbb{E}_{\boldsymbol{\pi}} \left[ r^k \left( s_t, a_t^k, a_t^{-k}, \boldsymbol{\mu} \right) - r^k \left( s_t, a_t^k, a_t^{-k}, \boldsymbol{\mu}' \right) \right] | \leq \mathcal{C} | \boldsymbol{\mu} - \boldsymbol{\mu}' | \tag{5}$$

Where $\mathcal{C} = (1 - \gamma)^{-1} C$. The last equation is drawn form the Assumption 1 and the inequality of a geometric series: $| (I - \gamma p^{\pi})^{-1} | \leq (1 - \gamma)^{-1}$. Some parts follow the same logic of (Bacon et al., 2017; Mguni et al., 2019; Kakade & Langford, 2002). □

**Proposition 2.** *(Leader-Follower Consistency.) If both Assumptions 1 and 2 are satisfied, for every $\epsilon > 0, k \in \mathcal{N}$, there exists $\delta > 0$, such that $| \boldsymbol{\mu} - \boldsymbol{\mu}' | \leq \epsilon$ implies $\left| \pi^k - \pi'^k \right| \leq \delta$, where $\boldsymbol{\mu}'$ and $\pi'^k$ are the new policies for the leader and follower $k$ respectively.*

*Proof.* By combining Lemma 2 and Assumption 2, we can draw that:

$$\left| \pi^k - \pi'^k \right| \leq (1 - \gamma) C | \boldsymbol{\mu} - \boldsymbol{\mu}' |$$

If there exist $| \omega - \omega' | < \epsilon$ and we have:

$$\left| \pi^k - \pi'^k \right| \leq (1 - \gamma) C | \boldsymbol{\mu} - \boldsymbol{\mu}' | \leq \epsilon (1 - \gamma) C$$

And we set $\delta \geq \epsilon (1 - \gamma) C$, the consistency is established. □

**Lemma 3.** *(Action Abstraction Policy Gradient.) Under the assumption that the low-level follower policy $\pi_{lower}^k (a_t | s_t, z_t)$ is fixed and deterministic, the policy gradient for action abstraction-based follower can be formulated as:*

$$\nabla_{\lambda^k} J^k \propto \mathbb{E} \left[ \nabla_{\lambda_k} \log \pi_{meta}^k (z | \hat{s}) R^k \right],$$

*Where $R^k$ is the accumulated reward for the $k$-th follower.*

*Proof.* Recall the utility for the $k$-th follower:

$$J^k (\boldsymbol{\mu}, \boldsymbol{\pi}) = \mathbb{E} \left[ \sum_{t=0}^{T} \gamma^t r_t^k \left( s_t, \boldsymbol{a}_t, \boldsymbol{\omega}_t \right) | \boldsymbol{a}_t \sim \boldsymbol{\pi} \left( \cdot | s_t \right), s_{t+1} \sim P \left( \cdot | s_t, \boldsymbol{a}_t \right), s_0 = s, \boldsymbol{\omega}_t \sim \boldsymbol{\mu} (\cdot | s_t, \boldsymbol{\omega}_{t-1}) \right]$$

For brevity, we rewrite the object as: $J^k = \mathbb{E} \left[ \sum_{t=0}^{T} \gamma^t r_t^k \left( s_t, \boldsymbol{a_t}, \boldsymbol{\omega}_t \right) \right]$ with a slight abuse of notation. The standard policy gradient can be:

$$\nabla_{\lambda^k} J^k = \mathbb{E} [ \nabla_{\lambda^k} \log \pi_t^k (a_t^k | s_t) R^k ]$$

When $\pi_{lower}^k(a|\hat{s}, z)$ is fixed and deterministic, the equation can be:

$$\nabla_{\lambda^k} J^k = \mathbb{E}\left[\nabla_{\lambda^k} \log[\sum_{z_t} \pi_{meta}^k(z_t|\hat{s}_t)\pi_{lower}^k(a_t|\hat{s}_t, z_t)]R^k\right] = \mathbb{E}\left[\frac{\nabla_{\lambda^k}[\sum_{z_t} \pi_{meta}^k(z_t|s_t)\pi_{lower}^k(a_t|\hat{s}_t, z_t)]}{\sum_{z_t} \pi_{meta}^k(z_t|\hat{s}_t)\pi_{lower}^k(a_t|\hat{s}_t, z_t)}R^k\right]$$

$$= \mathbb{E}\left[\frac{\nabla_{\lambda^k}[\sum_{z_t} \pi_{meta}^k(z_t|\hat{s}_t)\pi_{lower}^k(a_t|\hat{s}_t, z_t)]}{\sum_{z_t} \pi_{meta}^k(z_t|\hat{s}_t)\pi_{lower}^k(a_t|\hat{s}_t, z_t)}R^k\right] = \mathbb{E}\left[\frac{[\sum_{z_t} \nabla_{\lambda^k}\pi_{meta}^k(z_t|\hat{s}_t)\pi_{lower}^k(a_t|\hat{s}_t, z_t)]}{\sum_{z_t} \pi_{meta}^k(z_t|\hat{s}_t)\pi_{lower}^k(a_t|\hat{s}_t, z_t)}R^k\right]$$

$$= \mathbb{E}\left[\frac{[\sum_{z_t} \nabla_{\lambda^k}\pi_{meta}^k(z_t|\hat{s}_t)\pi_{lower}^k(a_t|\hat{s}_t, z_t)]}{Z}R^k\right]$$

$$\propto \mathbb{E}\left[\nabla_{\lambda^k} \log[\pi_{meta}^k(z_t|\hat{s}_t)] R^k\right]$$

Where $Z = \sum_z \pi_{meta}^k(z_t|\hat{s}_t)\pi_{lower}^k(a_t|\hat{s}_t, z_t) = 1$ is the partition function. For brevity, with a slight abuse of notation, we omit the superscript for variables $a$ and $z$ which represents the index of an agent.

□

# B    LOSS FUNCTIONS

## B.1    LEADER LOSS FUNCTIONS

We add a baseline function to reduce the variance of the event-based policy gradient for the leader. We adopt the idea of successor representation (Rabinowitz et al., 2018; Shu & Tian, 2019) as two expected baseline functions: $\hat{\phi}^g(c_t)$ and $\hat{\phi}^b(c_t)$. For the gain baseline function:

$$\phi^g = \sum_{g \in \mathcal{G}} \sum_{k \in \mathcal{N}} \mathbb{I}\left(g = g_t^k\right) \mathbb{I}\left(s_t^k = s_g\right) v_g,$$

For the bonus-based baseline function:

$$\phi^b = -\sum_{g \in \mathcal{G}} \sum_{k \in \mathcal{N}} \mathbb{I}\left(g = g_t^k\right) b_t^k,$$

Two baseline neural network functions with parameters $\hat{\theta}_g$ and $\hat{\theta}_b$ are trained through minimizing the mean square error:

$$L_{baseline} = (\phi^g(c_t; \hat{\theta}_g) - \phi^g)^2 + (\phi^b(c_t; \hat{\theta}_b) - \phi^b)^2,$$

Where $c_t$ is the attention-based latent variable.

To this end, the gradient for the leader can be formulated as:

$$\nabla_\theta L_l^{policy} = -\nabla_\theta J'(\theta) + \nabla_\vartheta(\lambda_1 L_{reg}); \quad \nabla_\vartheta L_l^{policy} = -\nabla_\vartheta J'(\vartheta) + \nabla_\vartheta(\lambda_2 L_{enp}).$$

Where

$$\nabla_\theta J'(\theta) \approx \mathbb{E}_{\tau \sim p_\tau(\cdot)}\left\{\left[\sum_{k \in \mathcal{N}} \sum_t I(e_t^k)\right](R^\tau(T) - \phi^g(c_t; \hat{\theta}_g) + \phi^b(c_t; \hat{\theta}_b))\right\},$$

$$\nabla_\vartheta J'(\vartheta) \approx \mathbb{E}_{\tau \sim p_\tau(\cdot)}\left\{\left[\sum_{k \in \mathcal{N}} \sum_t I^1(e_t^k)\right](R^\tau(T) - \phi^g(c_t; \hat{\theta}_g) + \phi^b(c_t; \hat{\theta}_b))\right\},$$

are the baseline policy gradients.

We also leverage the imitation learning to learn the action probability function $p^k\left(a_t^k|s_t, h_t^k, \hat{\theta}_I\right)$ similar to (Shu & Tian, 2019), where $\theta_I$ is the parameters for follower-aware module:

$$L_{\mathrm{IL}} = \mathbb{E}\left[-\frac{1}{N} \sum_{k \in \mathcal{N}} \log p^k\left(a_t^k|s_t, h_t^k; \hat{\theta}_I\right)\right].$$

The history encoder, the state encoder and the attention module are updated with the leader's policy gradient end-to-end.

## B.2 Follower Loss Functions

The follower policy gradient is from Lemma 3 and we find that adding the history $h^k$ can improve the performance:

$$\nabla_{\lambda^k} J^k \propto \mathbb{E} \left[ \nabla_{\lambda_k} \log \pi^k_{meta}(z|s, h^k) R^k \right],$$

and the learning rate of the follower $\alpha$ and the leader $\beta$ satisfy

$$\sum_{t \geq 0} \alpha_t = \sum_{t \geq 0} \beta_t = \infty, \quad \sum_{t \geq 0} \alpha_t^2 + \beta_t^2 < \infty, \quad \lim_{t \to \infty} \beta_t/\alpha_t = 0.$$

Which indicates the follower's learning rate is much higher than the leader. $\hat{\beta}$ is the learning rate for the leader's critic function, which is the same as $\alpha$. Followers do not have critic function.

## C Algorithm

---

**Algorithm 1:** EBPG

---

**Input:** The initialized leader's parameters $\theta$, $\vartheta$ and followers' parameters $\lambda$;
**Output:** Well-trained leader and followers;

1 **while** *not converge* **do**
2     **if** *Rollout Stage* **then**
3         **for** $t \leq T$ **do**
4             leader commits to the goals $g$ and bonuses $b$ according to Algorithm 2;
5             send the goals and bonuses to each agent separately;
6             **for** $k$ *in* $N$ **do**
7                 The $k$-th follower receives its own bonus $b_t^k$ and goal $g_t^k$;
8                 each agent make a decision $a_t^k = \pi_t^k(a_t^k|\hat{s}_t)$;
9             **end**
10             $s_{t+1}, r_t^l, \{r_t^k\}_{k \in \mathcal{N}} = step(s_t, \boldsymbol{a}_t)$;      // transition function.
11         **end**
12         store $\langle s_t, \boldsymbol{a}_t, \boldsymbol{\omega}_t, s_{t+1}, r_t^l, \{r_t^k\}_{k \in \mathcal{N}} \rangle$ in episode buffer;
13     **else if** *Training Stage* **then**
14         **for** $k \in \mathcal{N}$ **do**
15             $\lambda^k \leftarrow \lambda^k - \alpha \nabla_{\lambda^k} J^k$;      // the follower's parameters
16         **end**
17         $\hat{\theta}_g \leftarrow \hat{\theta}_g - \hat{\beta} \nabla_{\hat{\theta}_g} L_{baseline}, \hat{\theta}_b \leftarrow \hat{\theta}_b - \hat{\beta} \nabla_{\hat{\theta}_b} L_{baseline}$;    // the critic's parameters
18         $\hat{\theta}_I \leftarrow \hat{\theta}_I - \beta L_{\text{IL}} \nabla_{\hat{\theta}_I} L_{\text{IL}}$;   // the follower-aware module's parameters
19         $\theta \leftarrow \theta - \beta \nabla_\theta L_l^{policy}$;    // the termination function's parameters
20         $\vartheta \leftarrow \vartheta - \beta \nabla_\vartheta L_l^{policy}$;    // the parameters of leader's policy;
21 **end**

---

**Algorithm 2:** Action Choices for Leader

---

**Input:** Leader's policy $\boldsymbol{\mu}$
**Output:** goals $\boldsymbol{g}$ and bonuses $\boldsymbol{b}$;

1 **for** $k \in \mathcal{N}$ **do**
2     sample the termination function $\beta_t^k$;
3     **if** $\beta_t^k$ *terminates previous leader's action* $\omega_{t-1}^k$ **then**
4         choose new action $\omega_t^k$;
5     **else**
6         $\omega_t^k = \omega_{t-1}^k$;      // maintain previous action.

---

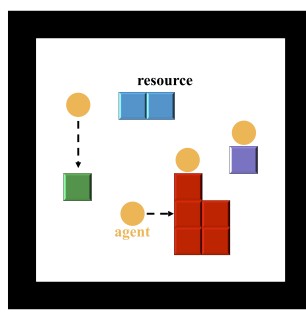 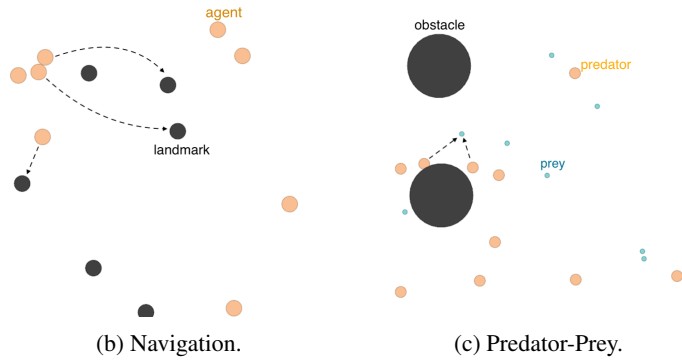

(a) Resource Collections & Multi-bonus Resource Collections. This figure is inspired by (Shu & Tian, 2019).

(b) Navigation.

(c) Predator-Prey.

Figure 5: Illustration of different tasks.

# D  EXPERIMENT DETAILS

## D.1  TASKS DETAILS

The illustration of experimental scenarios can be found in Figure 5. Here we give some details about these environments:

*Resource Collections.* This task is similar to (Shu & Tian, 2019), which is based on the scene that the leader and the followers collect some resources. Each follower has its own preference which might be the same (or against) to the leader's preference. In order to make the followers obey the leader's instruction, the leader should pay the followers bonuses. There are total 4 types of resources and for different resources each agent has different preferences. The leader owns two type of bonus (1 or 2) and 4 types of goals (each resource is a goal). The number of leader is 1 while the number of followers is 4.

*Multi-Bonus Resource Collections.* This task is similar to *Resource Collections*. Except that the leader can take 4 level bonuses (a bonus from 1 to 4) while each agent owns one skill. The number of leader is 1 while the number of followers is 4.

*Modified Navigation.* This task is original from (Lowe et al., 2017). We make some modifications here to make it suitable our SMG: the leader and the followers are going to navigate some landmarks. Each follower has its own preference which might be the same (or against) to the leader's preference. When a landmark has been navigated, it disappears immediately and a new landmark will appear. There are total 6 types of landmarks and for different landmarks, each agent has different preferences. The leader owns two type of bonuses and 6 types of goals (each landmark is a goal). The number of leader is 1 while the number of followers is 8 and the number of the landmarks is 6.

*Modified Predator-Prey.* This task is also original from (Lowe et al., 2017). We make some modification here to make it suitable our SMG: the leader and the followers are going to catch some preys. Each follower has its own preference which might be the same (or against) to the leader's preference. In each step, whether a prey has been caught, it randomly chooses a direction to go. Catching a prey will not make it disappear, which means that the preys can exist until the game ends. There are total 8 types of preys and for different preys, each agent has different preferences. The leader owns two type of bonuses and 8 types of goals (each prey is a goal). The followers are *faster* than the preys. The number of leader is 1 while the number of followers is 10 and the number of the landmarks is 8.

**Reward Design.** The rewards mentioned in Section 3 are the general forms. Here, we define two specified forms of the leader and followers reward function in our experiments:

*Leader Reward.* We define $v_g$ as the prize (utility) for finishing task $g$. We set the reward function for the leader at step $t$ as: $r_t^l = \sum_{g \in \mathcal{G}} \sum_{k \in \mathcal{N}} \mathbb{I}\left(g = g_t^k\right) \left(\mathbb{I}\left(s_t^k = s_g\right) v_g - b_t^k\right)$, formulated by the total gain $\sum_{g \in \mathcal{G}} \sum_{k \in \mathcal{N}} \mathbb{I}\left(g = g_t^k\right) \mathbb{I}\left(s_t^k = s_g\right) v_g$ (the total prizes got by the leader at time $t$) minus the total payment $-\sum_{g \in \mathcal{G}} \sum_{k \in \mathcal{N}} \mathbb{I}\left(g = g_t^k\right) b_t^k$ (the total bonuses paid to the followers). We should

emphasize that our leader reward is total different from the (Shu & Tian, 2019): in their approaches, the leader changes its mind after signing a contract will not be punished. To make it suitable to the real world, we modify the reward as the leader should pay the followers bonuses immediately after signing the contract and cannot get back if it gives up the contract.

*Follower Reward.* For the followers, we set the reward for the $k$-th follower as: $r_t^k = \sum_g r_{g,t}^k = \left( u_{k,g}\, \mathbb{I}\left(s_t^k = s_g\right) + \mathbb{I}\left(g_t^l = g_t^k\right) \times b_t^k \right)$, where $u_{k,g}$ reveals the payoff of the $k$-th follower when finishing task $g$ (the preference). Specifically, $r_t^k$ indicates that the follower can either follow the leader's instruction or just do what it prefers to. The followers will receive reward immediately after signing the contract (the leader and the followers achieve an agreement). $\mathbb{I}\left(s_t^k = s_g\right)$ means that the follower finishes the task $g$ at step $t$. A penalty is added to the followers if the followers betray the leader (the followers and the leader sign the contract but the followers do not agree to work).

## D.2 Training Details

### D.2.1 Network Design

Our network is based on (Shu & Tian, 2019). Some do not suit our method. We do some modification here: (1) We change the vanilla attention mechanism (sum/average all the history and action of each follower together) to a follower-specified one: each follower has a weight which indicates how important the follower is at the current step. (2) The output for $g$ and $b$ are changed into the sequential form, i.e., we first calculate $p(g_t^k|c_t^k)$ to get $g_t^k$, then based on $g_t^k$, we choose $p(b_t^k|g_t^k, c_t^k)$. (3) The history information $h_t^k$ is compressed by the LSTM (Gers et al., 2000).

**Leader's Network.** For the history information, we are willing to stress that the history consists of two parts, one is the statistical information similar to (Shu & Tian, 2019) and the other is the past information given by the neural networks. We leverage the LSTM with 128 hidden units and two fully-connected layers to compress the past information and the statistical information and obtain $h_t^k$. Each layer contains 128 neurons.

For the state information, we encode it into a 128-dimension hidden state by a state encoder. For resource collections task, the state encoder consist of a convolutional layer with 64 channels and kernels of $1 \times 1$ and two fully connected layers with 128 neurons. For navigation and predator-prey task, we encode the state using two fully connected layers with 128 neurons.

For the follower-specific attention module, we firstly predict the $\hat{a}_t^k$ from the follower-aware module which contains an LSTM with 128 hidden units and two fully-connected layers with 128 neurons. Secondly, we transform $\hat{a}_t^k$, $h_t^k$ and the output of the state encoder into $A(s_t, \hat{a}_t^k, h_t^k)$ using a fully connected layer with 128 neurons. The hidden vector is generated by concatenating all the $A(s_t, \hat{a}_t^k, h_t^k)$. Then the attention weight $w$ is calculated using two fully connected layers with softmax activation. The $\mathbf{c}_t$ is calculated as a weighted summation of $w$ and $A(\cdot)$. Finally, we obtain the $k$-th attention value $c_t^k$ by concatenating $\mathbf{c}_t$ and $A(s_t, \hat{a}_t^k, h_t^k)$.

For sequential graph-based model, we firstly calculate $p(g_t^k|c_t^k)$ using two fully connected layers. Each layer consists of 128 neuron units. The input of layers is $c_t^k$ and the output is $g_t^k$. Then we obtain $b_t^k$ by two fully connected layers. The input of layers is the concatenation of $c_t^k$ and $g_t^k$.

**Follower's Network.** We construct the RL-based followers by a state encoder and two fully connected layers. Each fully connected layer contains 128 neurons. For resource collections task, the state encoder consists of a convolutional layer with 64 channels and kernels of $1 \times 1$, two fully connected layers with 128 neurons and an LSTM with 128 hidden units. For navigation and predator-prey task, we encode the state using two fully connected layers with 128 neurons and an LSTM with 128 hidden units. Notice that the network structure of action-abstraction RL-based follower and non-action-abstraction RL-based follower is similar. The only difference is that the output of the former is high-level action $z$ and the output of the latter is low-level action $a_t^k$.

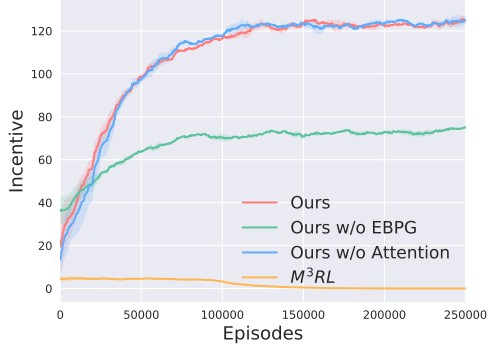

(a) Total incentive for the predator-prey task (rule-based agent).

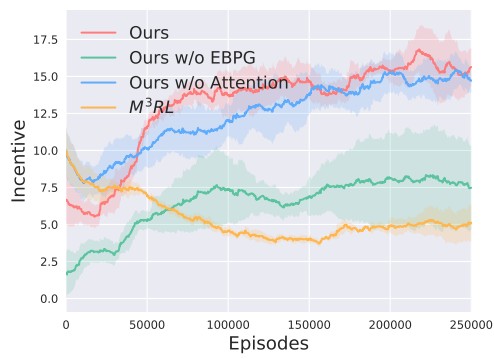

(b) Total incentive for the multi-bonus task (rule-based agent).

Figure 6: Total incentives for predator-prey task and multi-bonus resource collections task.

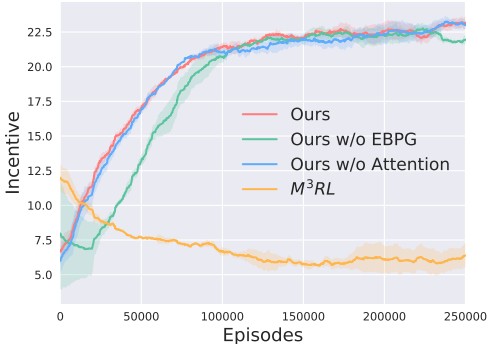

(a) Total incentive for the resource collections task (rule-based agent).

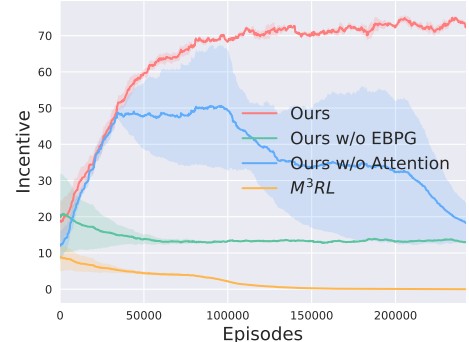

(b) Total incentives for the navigation task (rule-based agent).

Figure 7: Total incentives for resource collections task and navigation task.

### D.3 MORE EXPERIMENTAL RESULTS

#### D.3.1 TOTAL INCENTIVE FUNCTION

Total incentive (income) functions $R^{in} = \sum_t r_t^l + \sum_t \sum_k b_t^k$ can reveal how well the leader interacts with the followers; the higher the $R^{in}$ is, the more successful the coordination between the leader and the followers.

From Figure 6 and 7, comparing with the state-of-the-art method, we can see that our method far outperforms the state-of-the-art method, which reveals that our method does have a better ability to coordinate with the followers. In all of the scenarios, without the EBPG, the performance of our method is worse than ours with EBPG. Specifically, in some scenarios (e.g., multi-bonus resource collections, navigation), without the EBPG, the performance of our method is (or nearly) similar to the performance of $M^3RL$, showing the effectiveness of our novel policy gradient. Moreover, we can notice that in navigation environment, without follower-specified attention, the performance of our method diminishes rapidly, which implies that in some scenarios, attention does play an important role.

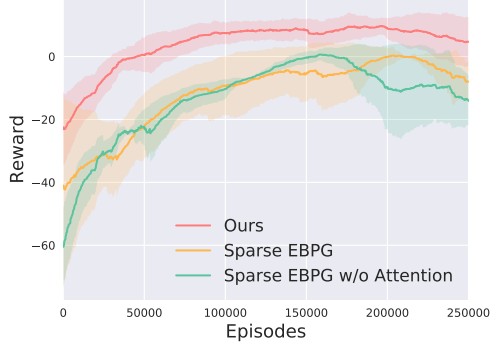

(a) The reward curves for multi-bonus resource collections.

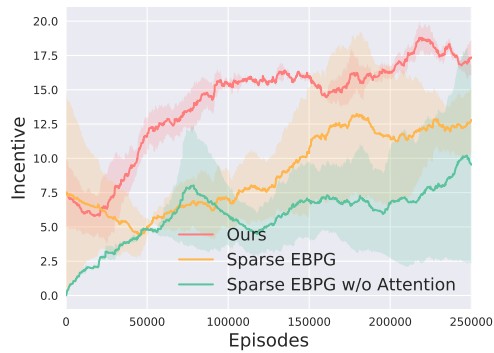

(b) The total incentive curves for multi-bonus resource collections.

Figure 8: The ablation study of sparse EBPG in the multi-bonus resource collections task.

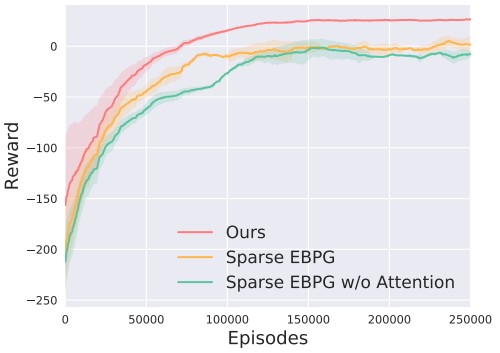

(a) The reward curves for predator-prey.

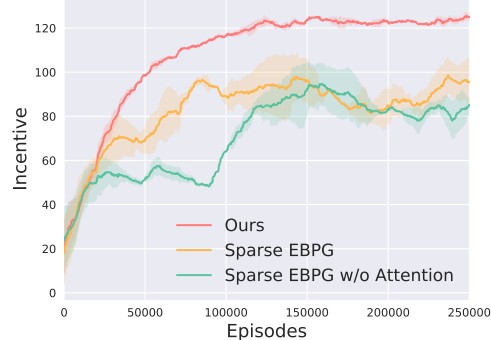

(b) The total incentive curves for predator-prey.

Figure 9: The ablation study of sparse EBPG in the predator-prey task.

### D.3.2 ROBUST OF OUR METHOD

In this section, we are going to testify the robust of our method. Specifically, we evaluate whether our method is robust to the noise. We make this experiment by introducing noise into the follower decision. For example, if we set the noise function as $30\%$, indicating that there is $30\%$ probability that the followers will choose action randomly.

### D.3.3 SPARSE EVENT-BASED POLICY GRADIENT

This experiment testifies whether the dense event-based policy gradient increases the leader's performance comparing with the sparse event-based policy gradient. We make ablations here: (1) Ours: the full structure of our method; (2) Sparse Event-Based Policy Gradient (sparse EBPG): the fully structure of ours except that the EBPG is replaced by sparse event-based policy gradient; (3) sparse EBPG w/o attention: replacing the follower-specified attention mechanism by averaging the input features.

From Figure 8 and 9 we can find that if the policy gradient is sparse, its performance is worse than the dense one, implying that the dense method does improve the leader's performance. There is also an interesting phenomenon that sparse EBPG with follower-specified attention mechanism performs better than that without, revealing that the attention can stabilize training when the training signal is sparse.

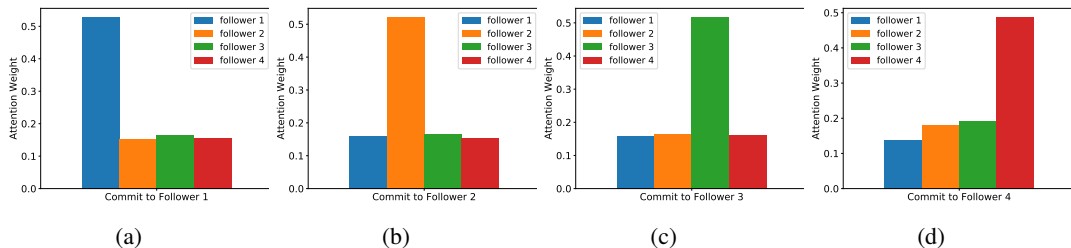

Figure 10: Visualization of the attention. The x-axis indicates the commit time for different agents and the y-axis indicates the corresponding weights $w^k$ when the leader commits to that agent.

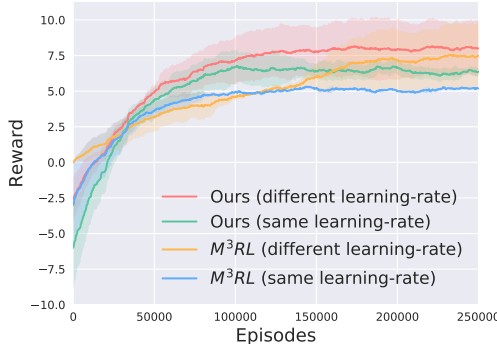

(a) The ablation study of reward curves for two time-scale update method (one RL-based follower).

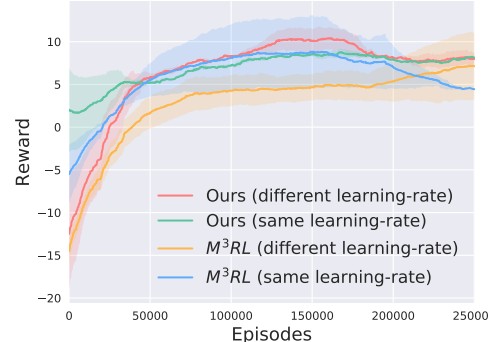

(b) The ablation study of reward curves for two time-scale update method (two RL-based followers).

Figure 11: The ablation study of reward curves for two time-scale update method in resource collections task.

### D.3.4 VISUALIZING ATTENTION

Following the same logic of (Iqbal & Sha, 2019), we visualize the weight of the attention when the leader takes actions. From Figure 10, we find that the attention mechanism does learn to strongly attend to the followers that the leader needs to take actions. The followers with leader's commitment obtain much higher attention weight than others, showing that the attention module actually learn to identify the important followers while leader committing new action. Thus, the attention mechanism does play an important role in improving the performance.

### D.3.5 TWO TIME-SCALE UPDATING

In order to evaluate the performance of our two time-scale update scheme (TTSU), we do an ablation study as shown in Fig 11. We can find that the performance where the followers' learning rate $\alpha$ ($1 \times 10^{-3}$) is much larger than the leader's $\beta$ ($3 \times 10^{-4}$) is better than the performance where the leader's learning rate is similar to the followers ($1 \times 10^{-3}$). Moreover, without TTSU, the reward curves of training methods become unstable, revealing that TTSU can stabilize the training process. In fact, TTSU improves the rate of convergence and play an important role in improving performance.

### D.3.6 COMMITTING INTERVAL

We evaluate the leader's performance between static committing interval and our dynamic committing interval. As shown in Figure 12, we observe that all the different fixed committing intervals only change the rate of convergence and do not enhance the leader's performance. All the fixed

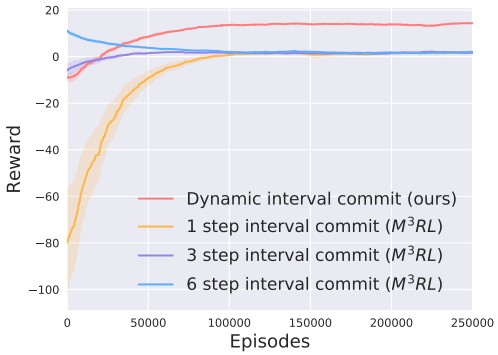

(a) The ablation study of reward curves for different committing interval in resource collections.

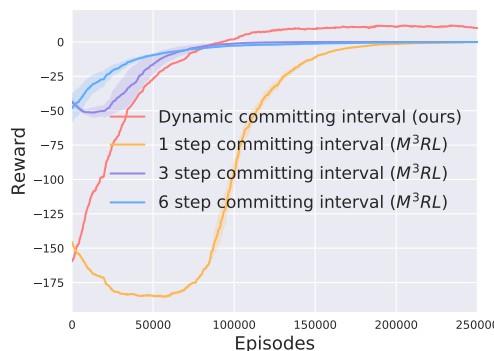

(b) The ablation study of reward curves for fixed committing interval method in navigation.

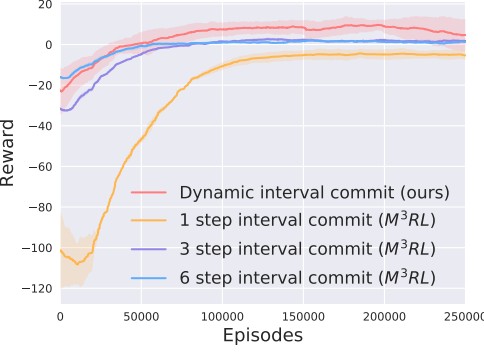

(c) The ablation study of reward curves for fixed committing interval method in multi-bonus resource collections.

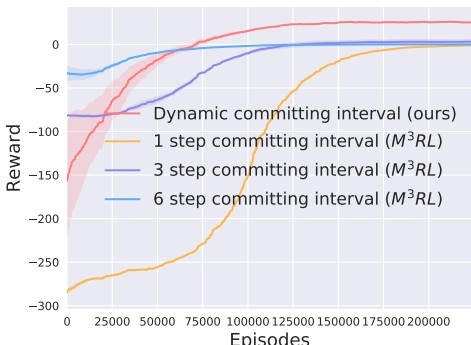

(d) The ablation study of reward curves for fixed committing interval method in predator-prey.

Figure 12: The ablation study of reward curves for fixed commitment time in resource collections task.

committing intervals are much worse than our dynamic committing approach, revealing the fact that our dynamic committing approach aids a lot in improving the leader's performance.

Table 2: Ablation results of RL-based followers for resource collections. '✓' means the module is used.

| Modules | | | | | | | | |
|---|---|---|---|---|---|---|---|---|
| EBPG | ✓ | ✓ | | | ✓ | ✓ | | |
| Leader-follower consistency | ✓ | | ✓ | | ✓ | | ✓ | |
| Action abstraction policy gradient | ✓ | ✓ | ✓ | ✓ | | | | |
| Final reward | 8.12 | 6.26 | 4.18 | 3.48 | -0.02* | -0.03* | -0.05* | -0.05* |

⋆ The last four terms are so close to zero and we use the rewards from last episode as the final results.

### D.3.7 REWARD FOR RL-BASED FOLLOWERS

We are interesting in the reward for the RL-based follower(s). Intuitively, a well-performing leader can make the follower gain more. As shown in Figure 13, the reward for RL follower is higher than M³RL follower in all the tasks. This represents the leader can coordinate the followers better and make them gain more reward than other methods, which forms a win-win strategy.

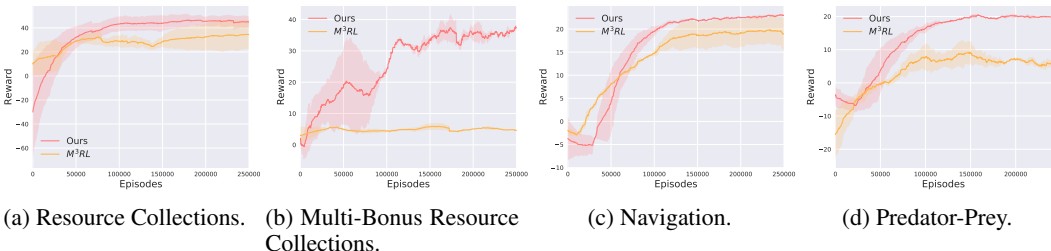

(a) Resource Collections.  (b) Multi-Bonus Resource Collections.  (c) Navigation.  (d) Predator-Prey.

Figure 13: Reward curves for RL-based followers in different tasks.

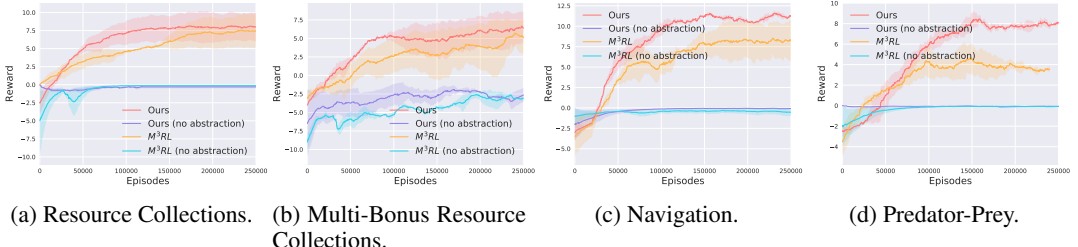

(a) Resource Collections.  (b) Multi-Bonus Resource Collections.  (c) Navigation.  (d) Predator-Prey.

Figure 14: Leader's reward curves for different tasks (RL-based followers).

### D.3.8 NUMBER OF THE FOLLOWER AGENTS

Finally, we evaluate the leader's performance with different number of RL-based followers. As shown in Figure 15, we find that our method outperforms the state-of-the-art method when facing different number of RL-based workers.

### D.3.9 DIFFERENT COMBINATIONS STUDY

To further illustrate the performance of different combinations of our methods, we make an extra ablation here. We choose the resource collections as the environments. Our analysis is as follows:

For the RL based followers scenario, As shown in Table 2, we find that the action abstraction policy gradient is very important to converge. Additionally, adding different modules can improve the performance and the method with all the modules reach the highest reward than other combinations. The contribution ranking for each module is: Action abstraction policy gradient > EBPG > Leader-follower consistency.

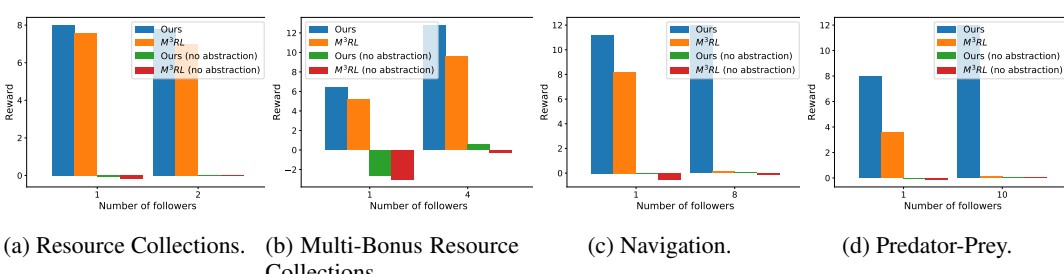

(a) Resource Collections.  (b) Multi-Bonus Resource Collections.  (c) Navigation.  (d) Predator-Prey.

Figure 15: The final reward of different number of follower agents in different tasks.

