# OpenReview forum: "Learning Expensive Coordination: An Event-Based Deep RL Approach"
_ICLR.cc/2020/Conference — Accept (Poster)_

### Official Review · AnonReviewer3 · 2019-10-22
**Official Blind Review #3**

**Rating:** 6

**Review:**

This paper proposes an interesting event-based policy gradient algorithm for single-leader multi-follower Stackelberg Markov Games (SMG). The authors claimed 3 contributions on their work: model the leader's decision-making process as a semi-MDP and proposed an event-based policy gradient algorithm for the leader to consider long-term effect; exploit a leader-follower consistency scheme with a follower-aware module and a follower-specific attention module to better predict the follower's behaviors; and propose an action-abstraction policy gradient algorithm in order to make SMG easier to converge.

The proposed model is interesting and the authors have done sufficient empirical studies to show the success of their method, as well as some necessary theoretical derivations. My major concern about this work is whether the task that this paper follows is widely studied in the reinforcement learning community, as the authors mentioned that only [1] can be used as baseline methods since the other methods can not be applied in their problem. But in general I think this is an interesting paper. Some detailed comments and suggestions:

1. This paper proposed 3 major contributions: event-based policy gradient, leader-follower consistency scheme and action abstraction policy gradient. The authors show the empirical performance comparison for the case with and without each of these techniques in the experiments. It can be seen that all of the 3 techniques help the proposed method perform better. In addition, the proposed method performs better compared to the state-of-the-art method (in [1]). Hence the authors have shown the success of their algorithm empirically. One thing that can be done to improve this paper is to show a joint comparison for the 2 ^ 3 = 8 cases of with / without any of the 3 techniques so that we can gain a better understanding about the effect of these 3 techniques.

2. The authors provide some necessary theoretical derivations to compute the policy gradients. But there are some typos in the equations. For example, the numerator of the equation of the case e_i^k\not\in A_T in I(e_i^k) at the bottom of page 4 is incorrect. In addition, the equation on the second last line of page 3 should be P_\gamma(s_{t+1}, \omega_t, a_t | s_t, \omega_{t-1}), not P_\gamma(s_{t+1}, \omega_t | s_t, \omega_{t-1}, a_t).  Please check the details and solve these typos.

3. It will be better if the authors can provide more theoretical analysis or quantitative explanation about the proposed method. For example, why a dense EBPG better than a sparse one? Why the equations in the attention mechanism appear in their form, not some similar forms (e.g. scale the values from the function A before putting them into the softmax function)? Intuitively the proposed techniques make sense, but it will be great if more insights are provided.

4. As I mentioned, I am concerned if the proposed problem is widely studied in the reinforcement learning community. Also, since the authors mentioned in the appendix that two of the tasks are original from [2], but we can not apply the method in [2] as a baseline method. It is possible to modify the method in [2] to fit with the setting and compare with them?

5. It will be better if the authors can rearrange the paper for a little bit, since there are too many important details (e.g. experiment settings, assumptions in the theorems) are in appendix.

Questions:

1. Can the authors try to modify the method in [2] and compare with it (details in my comment 4)?

Reference:

[1] Tianmin Shu and Yuandong Tian. M3RL: Mind-aware multi-agent management reinforcement learning. In ICLR, 2019.

[2] Ryan Lowe, Yi Wu, Aviv Tamar, Jean Harb, OpenAI Pieter Abbeel, and Igor Mordatch. Multi-agent actor-critic for mixed cooperative-competitive environments. In NeurIPS, pp. 6379–6390, 2017.

**Experience Assessment:**

I have read many papers in this area.

**Review Assessment: Checking Correctness Of Derivations And Theory:**

I assessed the sensibility of the derivations and theory.

**Review Assessment: Checking Correctness Of Experiments:**

I assessed the sensibility of the experiments.

**Review Assessment: Thoroughness In Paper Reading:**

I read the paper at least twice and used my best judgement in assessing the paper.

---

> ### Author Response · Authors · 2019-11-14
> **Response to Reviewer 3 (Part 1/2)**
>
> Thank you very much for your detailed and insightful comments. Here are our responses to your questions (we would like to start with Question 4):
>
> Q4 (part 1/2):” Whether our problem is widely studied and the contributions to the RL community.”
>
> We would like to highlight that Stackelberg (leader-follower) Games (SGs) with RL solvers have been explored in Multi-Agent Systems (MAS) and RL community for decades (Tharakunnel et al., 2007; Mguni et al., 2019, Cheng et al., 2017, Sabbadin & Viet, 2013, Sabbadin & Viet, 2016, Laumonier & Chaib-draa, 2005). However, most of these methods only focus on simple matrix games or small-scale Markov games and it is very difficult to extend them to handle complicated (large-scale states) Stackelberg Markov games (the problem we solve) because the training process is hard to stabilize in those environments as shown in our experiments. This is the reason why only a few papers focus on these environments.
>
> The large-scale states/ actions SMGs are also significant because they can model some real-world problems better than the matrix games and the small-scale SMGs. That is, in our real world, many non-cooperative groups naturally change with time and with large-scale states, e.g., the order dispatching for a taxi company in a city where the environment is the whole city and the state and action change with time [3]. Other examples include the league and the soccer clubs as well as the manager and employees. Therefore, investigating the large-scale states/ actions SMGs is very important for many real-world applications.
>
> We believe that investigating these complicated SMGs can provide a novel solution for non-cooperative multi-agent coordination. Nowadays, most of the prevailing deep MARL methods assume that the agents selflessly cooperate to maximize the global rewards, which is not true in many real-world non-cooperative scenarios (e.g., the order dispatch for taxi, the league and the soccer clubs, as well as the manager and employees). Taking the order dispatch for the taxi fleet as an example, order dispatch is a well-known application scenario for MARL [3,5]. However, many existing methods only consider how to maximize the total income of the company by forcing the drivers to accept the order which they may not be willing to. The learned strategy might not be efficient because the unsatisfied drivers may choose to deviate. On the contrary, our proposed method realistically assumes that drivers are self-interested. Therefore, exploring the SMGs and their solutions are necessary to learn efficient policies in many real-world applications. We add more real-world examples in the Introduction Section, on page 1.
>
> Besides that, our proposed method might inspire other areas in RL community, including multi-role MARL (our method introduces a training scheme for different roles) as well as hierarchical RL (our EBPG is a novel policy gradient for temporal abstraction). We add more details in the Conclusion Remarks Section, on page 9. We really hope our method would shed light on efficient MARL algorithm design involving self-interested agents.
>
>
> Q4 (part 2/2):” Compare MADDPG.”
>
> MADDPG is not a valid baseline for our method due to the following reasons:
> 1. Our problem has two kinds of players: the leader and the followers, where the leader move first and the followers best-respond to the leader’s policy. Therefore, the leader and the follower cannot be trained simultaneously. However, in MADDPG, all players are trained simultaneously. Therefore, we cannot directly add the leader role into the MADDPG framework.
> 2. On the other hand, we can use MADDPG only to train the followers. The training scheme can be depicted as follows: given the leader’s policy, we train followers by MADDPG, then we update the leader’s policy. It is our leader with DDPG-based followers structure and the only change between our method and this structure is the followers’ strategies (from the action abstraction policy gradient to the deterministic policy gradient). We have tried this scheme (with the Gumbel-SoftMax trick, since we focus on the discrete action space) at the very beginning of this work, as well as using other methods to train followers such as Q-learning, vanilla policy gradient. However, this training scheme cannot converge in a mild time limit, due to the complicated interactions among leaders and followers. That is also the reason why we design our action abstraction policy gradient.
> 3. The environments in [2] are open-sourced and widely used in the MARL community. We make lots of modifications to make them suitable for our problem (e.g. we add different gains for finishing different tasks by different agents). The new environments in our work are totally different from those in [2], therefore, the method in [2] cannot be used in our new environments (Details can be found in the Section D.1, on page 19).

---

> > ### Author Response · Authors · 2019-11-14
> > **Response to Reviewer 3 (Part 2/2)**
> >
> > (Continued)
> >
> > Q1: “Joint comparison.”
> >
> > We add an extra experiment on the joint comparison of our proposed approach in Resource Collections environment in Sec. D.3.9.  The main results are in the table below:
> >
> > Modules
> > EBPG                                                     | +     | +     |       |        | +     | +      |          |         |
> > Leader-follower consistency            | +     |        | +    |        | +     |         | +       |         |
> > Action abstraction policy gradient  | +     | +     | +    | +     |        |         |          |          |
> > Final reward                                        | 8.12 |6.26 |4.18|3.48 |-0.02|-0.03 |-0.05 |-0.05 |
> > ‘+’ means the module is used. This experiment is done in the Resource Collections environment with RL-based followers.
> >
> > As shown in the table above, we find that the action abstraction policy gradient is very important to converge for the RL-based followers. Additionally, adding different modules can improve the performance and the method with all the modules reach the highest reward than other combinations. The contribution ranking for each module in is RL-based followers: Action abstraction policy gradient > EBPG > Leader-follower consistency.
> >
> > Q2:” in equations.”
> >
> > We are sorry for making these typos. We fix them and also check the rest of the paper carefully according to your suggestions. Many thanks for reviewing our paper so carefully.
> >
> >
> > Q3:” More insights for the proposed methods.”
> >
> > We add more explanations about why we design our network and training scheme, including more details about the assumptions (pages 5-6), the attention module (page 6), as well as the EBPG (page 5).
> >
> > Additionally, we would like to discuss more details about the attention module. We do not adopt the scaling factor into our module for two reasons: 1). Our method is inspired by (Chen et al., 2018), which also do not use the scaling factor in their attention module. 2). The scaling approach is mainly used for the scenario that there is a high dimension dot-product operator before the SoftMax layer, which may lead to extremely small gradients and the scaling factor is used to prevent large input [4]. But for our structure, we do not have dot-product operation before the SoftMax layer and our input dimension is also small (only 128*1). Therefore, it is not necessary to introduce the scaling factor (the same statement is also mentioned in [4]).
> >
> > To testify our statement, we do an extra experiment on Resource Collections (with rule-based followers) and the results reveal that the final leader’s rewards for the method with and without scaling factor are similar (15.20 v.s. 15.17). Therefore, the scaling factor may not be very helpful.
> >
> > Q5:” Rearrange the paper.”
> >
> > We rearrange the paper by moving the assumptions (now on page 5) and some environment settings (hyper-parameters section, now on page 8) to the main text. Thanks!
> >
> > Reference:
> >
> > [3] Lin, Kaixiang, et al. "Efficient large-scale fleet management via multi-agent deep reinforcement learning." In  SIGKDD, pp. 1774-1783, 2018.
> >
> > [4] Vaswani, Ashish, et al. "Attention is all you need." In NeurIPS,pp. 5998-6008, 2017.
> >
> > [5] Li, Minne, et al. "Efficient Ridesharing Order Dispatching with Mean Field Multi-Agent Reinforcement Learning." In WWW,  pp. 983-994, 2019.

---

### Official Review · AnonReviewer1 · 2019-10-22
**Official Blind Review #1**

**Rating:** 8

**Review:**

This paper aims to improve the performance of leader agents in leader-follower Stackelberg Markov games (SMG) by introducing architectural components. an event-aware policy gradient for the temporally abstracted level of the leader, and a follower abstraction technique that mitigates the impact of follower distributional shift. The authors show significant improvements over the state of the art on several problems, and also evaluate the importance of the different components of their approach with thorough ablations.

I am not particularly familiar with the literature on SMGs, but I recommend this paper for acceptance. The architectural components (follower-wise attention), dense policy gradient, and follower-level abstractions are well-motivated, and the evaluations are extremely thorough. The supplementary material is longer than the main paper, and almost half of it is additional results.

The paper could be improved with more discussion of the applications of SMGs, although this could be due to my relative unfamiliarity with the area. For example, this seems immediately relevant to hierarchical RL problems that involve learning to coordinate multiple skills where the optimal subpolicies are incompatible, e.g. multi-robot coordination (or multi-limbed single robot coordination) with collisions. More discussion of the concrete applications would be informative.

Although I am not intimately familiar with the research area of SMGs, I expect this work will have significant impact in the area on the basis of the thoroughness of the results alone.

**Experience Assessment:**

I do not know much about this area.

**Review Assessment: Checking Correctness Of Derivations And Theory:**

I did not assess the derivations or theory.

**Review Assessment: Checking Correctness Of Experiments:**

I did not assess the experiments.

**Review Assessment: Thoroughness In Paper Reading:**

I read the paper at least twice and used my best judgement in assessing the paper.

---

> ### Author Response · Authors · 2019-11-14
> **Response to Reviewer 1**
>
> Thank you very much for your helpful and valuable comments that would surely help to improve the paper. In the following we discuss the potential applications of SMGs.
>
> SMGs could be applied to many multi-agent scenarios with the leader(s) and the self-interested follower(s), e.g., the taxi company and drivers (Mguni et al., 2019), the league and the soccer clubs (Sabbadin & Viet, 2016), as well as the local government and farmers (Sabbadin & Viet, 2013). Therefore, exploring SMGs can help the researchers in related areas to design efficient real-world systems (e.g., the ridesharing system for a taxi company which considers each driver’s preference). It can also be applied to multi-robot coordination environment when the robots are self-interested. We add more real-world examples in the Introduction Section, on page 1.
>
> Moreover, SMGs provide a novel and significant coordination problem to the RL community: individual agents no longer follow the centralized leader completely, which is different from the commonly seen centralized training methods in Dec-POMDP. We add these statements in the Conclusion Remarks on page 9 to illustrate more details.
>
> In all, SMGs can model various non-cooperative systems and we believe it will aid us to design more well-performing systems in real-world applications.

---

### Official Review · AnonReviewer2 · 2019-10-23
**Official Blind Review #2**

**Rating:** 6

**Review:**

1. Summary

The authors apply MARL to principal-agent / mechanism design problems where selfish agents need to be incentivized to coordinate towards a leader's (collective) goal.

The leader is modeled as a semi-MDP with event-based policy gradients and modules to model/predict followers' actions. The leader sends messages to followers, an "event" is a pair (timestep, message of leader to a follower).
A `termination' menas that an agent should stop executing the previous selected action; the leader signals as such to the agent.
With this modeling step, the authors formulate an event-based policy gradient, which considers models for which goal to send to followers and when.

The authors compare this approach on 4 environments with M3RL, which also solves (extensions of) principal-agent problems.

2. Decision (accept or reject) with one or two key reasons for this choice.

Weak accept.

3. Supporting arguments

The approach seems sound and conceptually related to a multi-agent generalization of STRAW https://arxiv.org/pdf/1606.04695.pdf, where a planner predicts / commits to an action-plan for a single agent.

4. Additional feedback with the aim to improve the paper. Make it clear that these points are here to help, and not necessarily part of your decision assessment.

5. Questions

**Experience Assessment:**

I have published one or two papers in this area.

**Review Assessment: Checking Correctness Of Derivations And Theory:**

N/A

**Review Assessment: Checking Correctness Of Experiments:**

I assessed the sensibility of the experiments.

**Review Assessment: Thoroughness In Paper Reading:**

I made a quick assessment of this paper.

---

> ### Author Response · Authors · 2019-11-14
> **Response to Reviewer 2**
>
> Many thanks for your insightful and constructive advice for additional studies. Here are our responses to your comments:
>
> Comment_1: “Relationship between EBPG and STRAW.”
>
> We agree that the high-level idea of our method is related to STRAW [1]. Roughly speaking, STRAW is one of the hierarchical RL methods which maintains two modules: an action-plan module to generate a sequence of actions and a commitment module to decide when to terminate the macro-actions. Our structure has these similar components.
>
> However, there are several key differences between our method and STRAW. Specifically,
>
> 1). Our method focuses on the non-cooperative environment between the leader (similar to high-level policy in hierarchical RL) and followers (similar to low-level policy in hierarchical RL) with more than two agents. In our problem, the leader can be viewed as the high-level policy while the followers can be regarded as low-level policies. However, different from the commonly seen hierarchical RL, the low-level policies do not always follow the high-level policy. Since the low-level policies and the high-level policy are not always cooperative, they cannot be trained end to end, which are distinct from the STRAW and other hierarchical RL methods where the low-level policies follow the high-level policy completely and can be trained end to end. As shown in our experiments, without the follower’s action abstraction policy gradient and other modules (e.g., the attention-based module, the two-time scale training, as well as the sequential decisions module), it is very difficult (even impossible) to converge.
>
> 2). Our macro-action updating method is also different from STRAW. We introduce an event-based probability inference approach to design a novel policy gradient for both the leader’s policy as well as the termination function which is different from the A3C used in STRAW. Moreover, a successor-based value baseline function is further introduced to reduce the variance. Therefore, our method is different from STRAW in many aspects.
>
> Thank you for bringing this interesting work to our attention and we cite it in the Related Works section, on page 2.
>
> [1] Vezhnevets, Alexander, et al. "Strategic attentive writer for learning macro-actions." In NeurIPS, pp. 3486-3494, 2016.

---

### Author Response · Authors · 2019-11-14
**Response to all reviewers**

For all the reviewers:

We thank all the reviewers for their very helpful and insightful comments. We have uploaded a new version and summarize the major changes below. We also address the individual concerns in separate responses.

1.	We add more real-world examples to show more application scenarios of our approach in the Introduction section on page 1.
2.	We add more details in the Conclusion Remarks section (on page 9) to reveal the contributions to RL and multi-agent system communities.
3.	We add an extra experiment to show the performance of different combinations of our modules in Sec. D.3.9 (on page 25).
4.	We rearrange the paper by moving the assumptions (now on pages 5 and 6) and some environment settings (hyper-parameters section, now on page 8) to the main text.
5.	We fix some typos.

---

### Decision · Program_Chairs · 2019-12-19

**Decision:**

Accept (Poster)

**Comment:**

This paper tackles the challenge of incentivising selfish agents towards a collaborative goal. In doing so, the authors propose several new modules.

The reviewers commented on experiments being extremely thorough. One reviewer commented on a lack of ablation study of the 3 contributions, which was promptly provided by the authors. The proposed method is also supported by theoretical derivations. The contributions appear to be quite novel, significantly improving performance of the studied SMGs.

One reviewer mentioned the clarity being compromised by too much material being in the appendix, which has been addressed by the authors moving some main pieces of content to the main text.

Two reviewer commented on the relevance being lower because of the problem not being widely studied in RL. I would disagree with the reviewers on this aspect, it is great to have new problem brought to light and have fresh and novel results, rather than having yet another paper work on Atari. I also think that the authors in their rebuttal made the practical relevance of their problem setting sufficiently clear with several practical examples.